# Highly efficient and robust noble-metal free bifunctional water electrolysis catalyst achieved via complementary charge transfer

Nam Khen Oh[1,3], Jihyung Seo[1,3], Sangjin Lee[2,3], Hyung-Jin Kim[2], Ungsoo Kim[1], Junghyun Lee[1], Young-Kyu Han [2✉] & Hyesung Park[1✉]

The operating principle of conventional water electrolysis using heterogenous catalysts has been primarily focused on the unidirectional charge transfer within the heterostructure. Herein, multidirectional charge transfer concept has been adopted within heterostructured catalysts to develop an efficient and robust bifunctional water electrolysis catalyst, which comprises perovskite oxides ($La_{0.5}Sr_{0.5}CoO_{3-\delta}$, LSC) and potassium ion-bonded $MoSe_2$ (K-$MoSe_2$). The complementary charge transfer from LSC and K to $MoSe_2$ endows $MoSe_2$ with the electron-rich surface and increased electrical conductivity, which improves the hydrogen evolution reaction (HER) kinetics. Excellent oxygen evolution reaction (OER) kinetics of LSC/K-$MoSe_2$ is also achieved, surpassing that of the noble metal ($IrO_2$), attributed to the enhanced adsorption capability of surface-based oxygen intermediates of the heterostructure. Consequently, the water electrolysis efficiency of LSC/K-$MoSe_2$ exceeds the performance of the state-of-the-art Pt/C‖$IrO_2$ couple. Furthermore, LSC/K-$MoSe_2$ exhibits remarkable chronopotentiometric stability over 2,500 h under a high current density of 100 mA cm$^{-2}$.

[1] Department of Materials Science and Engineering, Perovtronics Research Center, Low Dimensional Carbon Materials Center, Ulsan National Institute of Science and Technology (UNIST), Ulsan, Republic of Korea. [2] Department of Energy and Materials Engineering and Advanced Energy and Electronic Materials Research Center, Dongguk University-Seoul, Seoul, Republic of Korea. [3] These authors contributed equally: Nam Khen Oh, Jihyung Seo, Sangjin Lee. ✉email: ykenergy@dongguk.edu; hspark@unist.ac.kr

Hydrogen is one of the cleanest and most sustainable energy sources that can provide high-energy-density fuel for electricity generation. Moreover, hydrogen production via electrocatalytic water splitting has been extensively explored recently[1,2]. Noble-metal-based Pt/Pd and $RuO_2/IrO_2$ electrocatalysts have been widely utilized in the cathodic hydrogen evolution reaction (HER) and anodic oxygen evolution reaction (OER), respectively, for efficient hydrogen generation from water electrolysis. Although noble-metal-based electrocatalysts exhibit excellent water electrolysis performance, these catalysts typically suffer from poor stability and scarcity in alkaline water electrolysis systems[3,4], which is undesirable in practical industrial applications. In addition, most noble-metal-based catalysts function optimally only for the half-cell reaction (either HER or OER) under specific electrolytic conditions[5]. Bifunctional catalysts that can be simultaneously used for both the cathode and anode reactions in the same electrolyte have been developed to improve water electrolysis performance and simplify the electrolysis system[6,7]. For bifunctional catalysts, the reaction kinetics and efficiency of the OER process can determine the reaction rate of overall water electrolysis due to its sluggish kinetics from the rigid double bond in O–O and multi-proton-coupled electron transfer steps[8,9]. Therefore, achieving effective OER catalytic activity in bifunctional catalysts is critical in maximizing water electrolysis performance.

To strategically enhance the water electrolysis efficiency of the bifunctional catalyst, heterostructured catalyst configurations, including transition-metal-based nanocrystals (e.g., Co, Ni, Fe, and Cu) or noble-metal-based composites (e.g., Pt, Ag, Pd, Au, and Rh), have often been investigated for their morphology control, compositional combination of the heterostructure, and foreign elemental doping[10–14]. Charge transfer within the heterostructured catalyst, which can modulate the electronic structure of the heterostructure and directly influence the Faradaic efficiency of the respective electrode[15,16], is critical in improving the efficiency of overall water electrolysis for bifunctional catalysts. Thus far, most bifunctional heterostructure-based catalysts have primarily utilized unidirectional charge transfer effects between heterostructure components[17,18], which can potentially limit an optimized electronic structure to achieve ideal HER and OER catalytic activities. Therefore, a different perspective on catalyst design is needed to effectively modulate the electronic structure of the catalyst such that its electrocatalytic activity is maximized in water electrolysis. Furthermore, most bifunctional catalysts demonstrate operational stability under relatively low current densities[19–23]. As the durability of the electrolytic catalyst can be severely degraded at high current densities, e.g., through the Ostwald ripening, aggregation, and detachment during the electrolytic reaction[24,25], robust electrochemical stability of the bifunctional catalyst should also be achieved for industrial consideration in addition to the high efficiency of water electrolysis.

In the present work, we have developed a heterostructured catalyst comprising perovskite oxide ($La_{0.5}Sr_{0.5}CoO_{3-\delta}$, LSC) and potassium ion-bonded molybdenum diselenide ($K-MoSe_2$) as the bifunctional catalysts for overall water electrolysis. The semiconducting 2H-phase $MoSe_2$ was moderately converted to metallic $1T-MoSe_2$ via charge transfer from potassium atoms during the potassium metal intercalation process[26]. In a previous report, Park et al. reported a heterostructured water electrolysis catalyst structure comprising perovskite oxide and transition-metal dichalcogenides (TMDs) featured with unidirectional charge transfer enabling the in-situ local phase transition in TMDs. Differently, the $LSC/K-MoSe_2$ system in this study characterizes the multidirectional charge transfer phenomenon, involving two-way charge transfer from K to $MoSe_2$ and from LSC to $MoSe_2$, which led to significantly improved water

electrolysis performance and operational stability. When $K-MoSe_2$ forms a heterostructure with LSC, the metallic-phase purity of $MoSe_2$ is significantly increased to over 90% through the complementary charge transfer from LSC and potassium atoms. The optimized $LSC/K-MoSe_2$ catalyst exhibits significantly enhanced HER and OER performance compared with those of LSC or $K-MoSe_2$, which is attributed to the increased electrical conductivity of $MoSe_2$ and improved oxygen intermediate adsorption in LSC. In particular, the OER catalytic activity of $LSC/K-MoSe_2$ outperforms that of the noble-metal $IrO_2$ catalyst in 1 M KOH. Consequently, the performance (e.g., overpotential and Tafel slope) and energy efficiency of overall water electrolysis of the $LSC/K-MoSe_2||LSC/K-MoSe_2$ couple surpass those of the state-of-the-art $Pt/C||IrO_2$ pair. Furthermore, the integrated overall water electrolysis exhibits excellent operational stability (over 2,500 h) without the decomposition of the catalyst under a high current density of $100\,mA\,cm^{-2}$.

## Results

**Morphology and elemental characteristics of $LSC/K-MoSe_2$.** An $LSC/K-MoSe_2$ heterostructure catalyst was synthesized via a simple and mass-producible solution process (Fig. 1a; see "Methods" section for details). In brief, LSC and $K-MoSe_2$ were synthesized via the sol-gel method and molten-metal-assisted intercalation[26], respectively; the as-prepared LSC and $K-MoSe_2$ were mixed via ball milling at specified weight percent ratios. The resulting $LSC/K-MoSe_2$ compounds prepared in powder and solution forms are shown in Fig. 1a. Morphological and elemental analyses were conducted via scanning electron microscopy (SEM) and transmission electron microscopy (TEM). Supplementary Fig. 1 shows the SEM image of the $LSC/K-MoSe_2$ heterostructure, indicating that $K-MoSe_2$ was uniformly adsorbed onto the LSC surface without aggregation, which can contribute to the high specific surface area of the heterostructure. Figure 1b shows the high-angle annular dark-field (HAADF) image of $LSC/K-MoSe_2$ and the corresponding TEM energy-dispersive spectroscopy (EDS) elemental mapping profile. All relevant components (La, Sr, Co, O, Mo, Se, and K) were uniformly distributed in the heterostructure, and the elemental compositions of each component of $LSC/K-MoSe_2$ obtained from the TEM–EDS spectrum are presented in Supplementary Fig. 2 with the prescribed elemental ratio. High-resolution TEM (HR-TEM) analysis was further performed to observe the atomic structure of $LSC/K-MoSe_2$ in detail. As shown in Fig. 1c–e, the coexistence of LSC and $K-MoSe_2$ can be clearly observed. LSC shows the well-known $ABO_3$ layered perovskite oxide structure comprising La and Sr at the A site, Co at the B site, and O at the anion site, and the fast Fourier transform (FFT) pattern indicated a highly crystalline structure corresponding to the (110) and (001) planes of the LSC in the [110] zone axis (Fig. 1d). $K-MoSe_2$ exhibits the typical 1T-phase atomic structure, and the FFT pattern on the [001] zone axis shows the desired crystalline structure of $K-MoSe_2$ corresponding to the (100) and (110) planes (Fig. 1e). The $K-MoSe_2$ flakes formed on the as-synthesized $LSC/K-MoSe_2$ are a few layers in thickness, indicating that the as-exfoliated $K-MoSe_2$ flakes were successfully integrated with the LSC without additional restacking (Supplementary Fig. 3).

**Electrochemical performance.** The electrochemical activity of $LSC/K-MoSe_2$ was investigated by analyzing the linear sweep voltammetry (LSV) curve for HER and OER, cyclic voltammograms, double-layer capacitance values, and electrochemical impedance spectroscopy (EIS) results. As shown in Supplementary Figs. 4–7, Supplementary Tables 1 and 2, various weight percent ratios of LSC and $K-MoSe_2$ were first evaluated, and the

optimal ratio of LSC/K-MoSe$_2$ (5:4) yielding the best electro-chemical catalytic performance was determined. This optimized configuration of LSC/K-MoSe$_2$ heterostructure was utilized for further analyses. The hydrogen generation activity of LSC/K-MoSe$_2$ was evaluated in the half-cell system, and Fig. 2a compares the polarization curves of LSC/K-MoSe$_2$, K-MoSe$_2$, LSC, and Pt/C in N$_2$-saturated 1 M KOH using a three-electrode system. The corresponding Tafel slope of HER in various electrode configurations derived from the obtained LSV profile is shown in Fig. 2b. The Pt/C catalyst exhibited an overpotential of 68 mV at 10 mA cm$^{-2}$ with a Tafel slope of 31 mV dec$^{-1}$, which suggests that the Volmer–Tafel reaction is a rate-determining step[27]. In the case of LSC/K-MoSe$_2$, the catalyst exhibited significantly higher HER activity than that of the LSC and K-MoSe$_2$. LSC/K-MoSe$_2$ showed an overpotential of 128 mV at 10 mA cm$^{-2}$ and a Tafel slope of 45 mV dec$^{-1}$, whereas K-MoSe$_2$ and LSC required 288 and 450 mV to reach 10 mA cm$^{-2}$ with Tafel slopes of 62 and 119 mV dec$^{-1}$, respectively. The considerably improved Tafel slope, and thus the HER performance, of LSC/K-MoSe$_2$ implies that the primary rate-determining step of the heterostructured catalyst becomes closer to the Volmer–Tafel pathway of noble metals[28,29]. The enhanced HER performance of LSC/K-MoSe$_2$ was also confirmed through charge transfer resistance ($R_{ct}$) analysis between the electrode and electrolyte, which can directly affect electrochemical performance. Figure 2c shows the Nyquist plots of LSC/K-MoSe$_2$, K-MoSe$_2$, and LSC for the HER obtained via EIS analysis, and the derived $R_{ct}$ values were 1.30, 2.56, and 5.78 Ω cm$^2$, respectively, which corroborates that charge transfer kinetics is most favorable for LSC/K-MoSe$_2$ than others. We also investigated the OER activity of various catalyst configurations in N$_2$-saturated 1 M KOH using a three-electrode system. Figure 2d shows the LSV curves of LSC/K-MoSe$_2$, K-MoSe$_2$, LSC, and IrO$_2$ for OER. The corresponding Tafel slope is shown in Fig. 2e. IrO$_2$, known as the best performing noble-metal-based catalyst in OER[5], showed an overpotential of 350 mV at 10 mA cm$^{-2}$ with a Tafel slope of 81 mV dec$^{-1}$. Notably, the OER catalytic performance of LSC/K-MoSe$_2$ exceeded that of IrO$_2$ with an overpotential as low as 230 mV to reach 10 mA cm$^{-2}$ and a Tafel slope of 79 mV dec$^{-1}$. In contrast, LSC required an overpotential of 420 mV to reach 10 mA cm$^{-2}$ with a Tafel slope of 131 mV dec$^{-1}$ and K-MoSe$_2$ exhibited negligible OER activity. It has been reported that the OER performance of perovskite oxide can be improved by the doping effect of molybdenum[30]. We thus conducted XPS analysis of Mo and Se in LSC/K-MoSe$_2$ under OER condition after water electrolysis to elucidate the potential doping effect of Mo on the OER performance of LSC/K-MoSe$_2$ (Supplementary Fig. 8 and Supplementary Table 3). The XPS results indicated partial oxidation of Mo and Se, but not the decomposition, in LSC/K-MoSe$_2$ in our proposed heterostructured electrocatalyst system. In general, molybdenum oxide is known to have negligible OER activity in an alkaline environment, which is consistent with the observation of poor OER performance of K-MoSe$_2$ in our study[31]. Thus, the origin of OER performance enhancement of LSC in the heterostructure is not attributed to the doping effect of molybdenum or its oxide derivatives. In our system, Mo, rather than serving as the main contributor for the OER active site, is believed to participate in the continuous charge transfer process with LSC during the OER reaction, which enhances the electrophilicity of LSC, leading to the increased adsorption of OER intermediates such as OH* and OOH*, and modulates the electronic structure of LSC/K-MoSe$_2$ favorable for OER process. The improvement in OER kinetics of LSC/K-MoSe$_2$ compared to that of LSC owing to the presence of Mo in the OER environment was also previously verified via experimental and computational analyses (low Tafel slope value (Fig. 2e), upshift of XPS spectra of Co 2p peak (Supplementary Fig. 14), and reduced

free energy barrier and density of states (Fig. 4f–h and Supplementary Figs. 21–23)). The $R_{ct}$ values of LSC/K-MoSe$_2$, K-MoSe$_2$, and LSC for OER obtained from the EIS analysis were 1.52, 74.0, and 1.91 Ω cm$^2$, respectively (Fig. 2f). The lower $R_{ct}$ in LSC/K-MoSe$_2$ than K-MoSe$_2$ and LSC indicates that the interfacial resistance between the electrode and electrolyte is reduced through heterostructure formation, contributing to the improved OER kinetics. These results suggest that the formation of the LSC/K-MoSe$_2$ heterostructure may induce potential physicochemical interactions between the LSC and K-MoSe$_2$, generating synergistic effects to lead to the observed excellent intrinsic catalytic activities for both the HER and OER processes. Below, we present detailed analyses on the heterogeneous composite structure to elucidate the origin of the enhanced catalytic performance.

**Analysis of physical properties of LSC/K-MoSe$_2$.** The crystal structures of 2H-MoSe$_2$, LSC/K-MoSe$_2$, K-MoSe$_2$, and LSC were investigated via X-ray diffraction (XRD). As shown in Fig. 3a, the preferred crystallographic orientation of 2H-MoSe$_2$ at (002) plane is observed at 13.8°[32], whereas that of K-MoSe$_2$ is blue shifted to 13.4° owing to the increased interlayer spacing between the MoSe$_2$ layers through the formation of potassium-intercalated MoSe$_2$. The preferred orientation of the (002) plane for K-MoSe$_2$ was consistently found at 13.4° for the LSC/K-MoSe$_2$ heterostructure. In addition, as shown in Supplementary Fig. 9, the crystal structures of K-MoSe$_2$ and LSC can be clearly identified from LSC/K-MoSe$_2$, indicating that each crystal structure of K-MoSe$_2$ and LSC is well preserved when forming the heterostructure through the ball mill process. Thermogravimetric analysis (TGA) was performed to examine the surface adsorption capability of the catalysts (Fig. 3b). Target materials, including LSC/K-MoSe$_2$, K-MoSe$_2$, and LSC, were preexposed to wet-air conditions to adsorb various environmental species in the atmosphere such as moisture, H, and OH groups. With incremental heat treatments up to 600 °C, LSC/K-MoSe$_2$ exhibited a weight loss of ~14%, whereas both K-MoSe$_2$ and LSC exhibited only a marginal weight loss of ~2%, demonstrating the improved surface adsorption capability of the heterostructure. Brunauer–Emmett–Teller (BET) analysis was conducted to investigate the specific surface area of LSC/K-MoSe$_2$ (Fig. 3c). Compared with K-MoSe$_2$ and LSC, the heterostructure exhibited significantly enhanced surface areas, i.e., 203.94, 104.51, and 32.54 m$^2$ g$^{-1}$ for LSC/K-MoSe$_2$, K-MoSe$_2$, and LSC, respectively, indicating that the active sites of the heterostructured catalyst can be increased to improve the efficiency of the electrochemical cell. We further analyzed the BET surface area and water electrolysis performance of various other conditioned samples comprising LSC and K-MoSe$_2$ to gain more insights for the surface area effect according to different ball-milling processes on the water electrolysis performance (Supplementary Figs. 10, 11 and Supplementary Table 4). We note that, overall, there was no significant difference in the morphology for each conditioned samples before and after the ball-milling process (Supplementary Fig. 12). For the individual component materials (LSC and MoSe$_2$), the ball-milled samples showed slightly increased BET surface areas, but the water electrolysis performance remained almost unaffected. When forming the heterostructured catalyst, the ball-milling process to the specific surface area and the electrolytic performance exhibited the most synergistic effects when applied during the heterostructure formation of individual component materials (i.e., LSC and K-MoSe$_2$) rather than to each component of LSC and K-MoSe$_2$ followed with the mixing process. These results indicate that ball milling during the formation of the composite structures increases the active sites for the water electrolysis and facilitates the charge transfer between LSC and K-MoSe$_2$, thereby

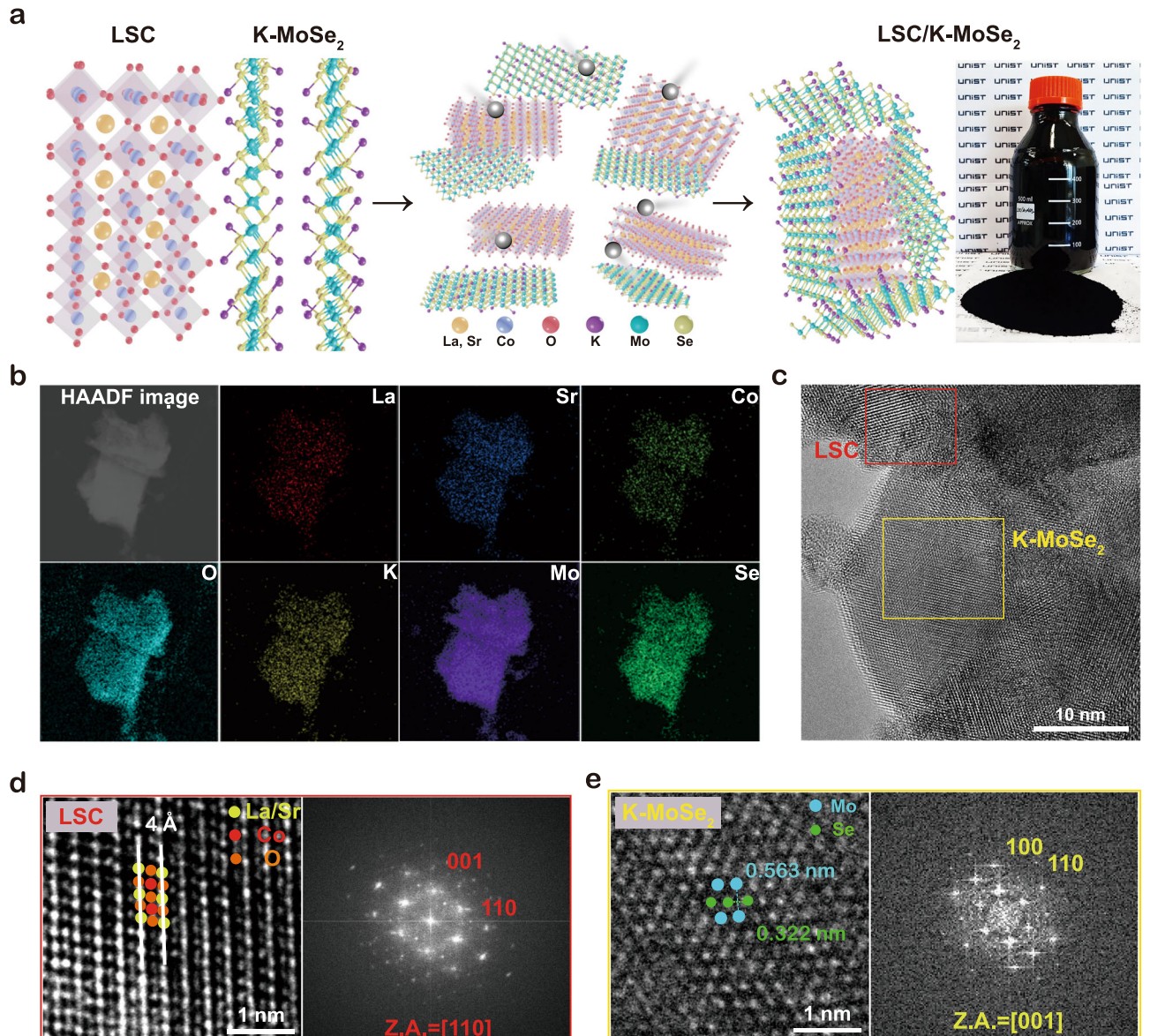

**Fig. 1 Elemental and morphological characterizations of LSC/K-MoSe₂ heterostructure. a** Schematic illustrating the synthesis process of LSC/K-MoSe₂. Digital image demonstrates the large-scale synthesis capability of the proposed method. **b** HAADF-STEM image and corresponding EDS elemental mapping of La (red), Sr (blue), Co (yellow green), O (cyan), K (yellow), Mo (purple), and Se (green). **c** HR-TEM image of LSC/K-MoSe₂, showing typical morphological characteristics of the heterostructure comprising the LSC (red square) and K-MoSe₂ (yellow square) regions. **d** Representative HR-TEM image of LSC and corresponding FFT image, indicating the lattice structure of La/Sr, Co, and O with high crystallinity. **e** Representative HR-TEM image of K-MoSe₂ and corresponding FFT image, exhibiting the highly crystalline lattice structure of 1T-phase MoSe₂ comprising Mo and Se.

enabling the improved water electrolysis performance in the heterostructured catalyst. In addition, the pore size of LSC/K-MoSe₂ was measured using the Barrett–Joyner–Halenda (BJH) method. LSC/K-MoSe₂ exhibited a mesoporous pore size distribution of 2–50 nm (Supplementary Fig. 13). The mesoporous nature of the catalyst can effectively improve the surface area of the catalyst as well as the diffusion of the electrolyte and ionic species[33]. Further, the desorption of hydrogen and oxygen generated during the overall water electrolysis reaction becomes favorable owing to the rapid mass transfer from the electrolyte to the catalyst surface, which can help to improve the performance of the water electrolysis[34,35].

Figure 3d shows the Raman spectra of LSC/K-MoSe₂ and K-MoSe₂. Characteristic vibrational Raman modes of K-MoSe₂ were detected at 106.2, 150.5, 221.6, 165.2, and 289.7 cm⁻¹, corresponding to the J₁, J₂, J₃, E₁g, and E¹₂g peaks of 1T-MoSe₂[36]. The characteristic

Raman peak positions of LSC/K-MoSe₂ were red shifted by 2 cm⁻¹, indicating potential electronic interaction between the LSC and K-MoSe₂[37]. Ultraviolet photoelectron spectroscopy (UPS) analysis was performed to investigate the charge transfer effect between LSC and K-MoSe₂. Work function values were derived from the secondary cutoff energies. Figure 3e shows that the work function of LSC/K-MoSe₂ increased compared to LSC and K-MoSe₂ alone (5.38 eV vs. 5.15 and 5.22 eV). This verifies that electronic structure modulation in LSC/K-MoSe₂ occurs via charge transfer between LSC and K-MoSe₂. Compared with LSC, LSC/K-MoSe₂ with an increased work function has several merits for improving electrochemical performance. The increased work function enhances the rate constant and preexponential kinetic factor in the electrocatalytic reaction[38,39]. The high rate constant and kinetic factor reduce the bond strength between active sites and adsorbed intermediates of HER on the catalyst surface, increasing the exchange current density and enabling

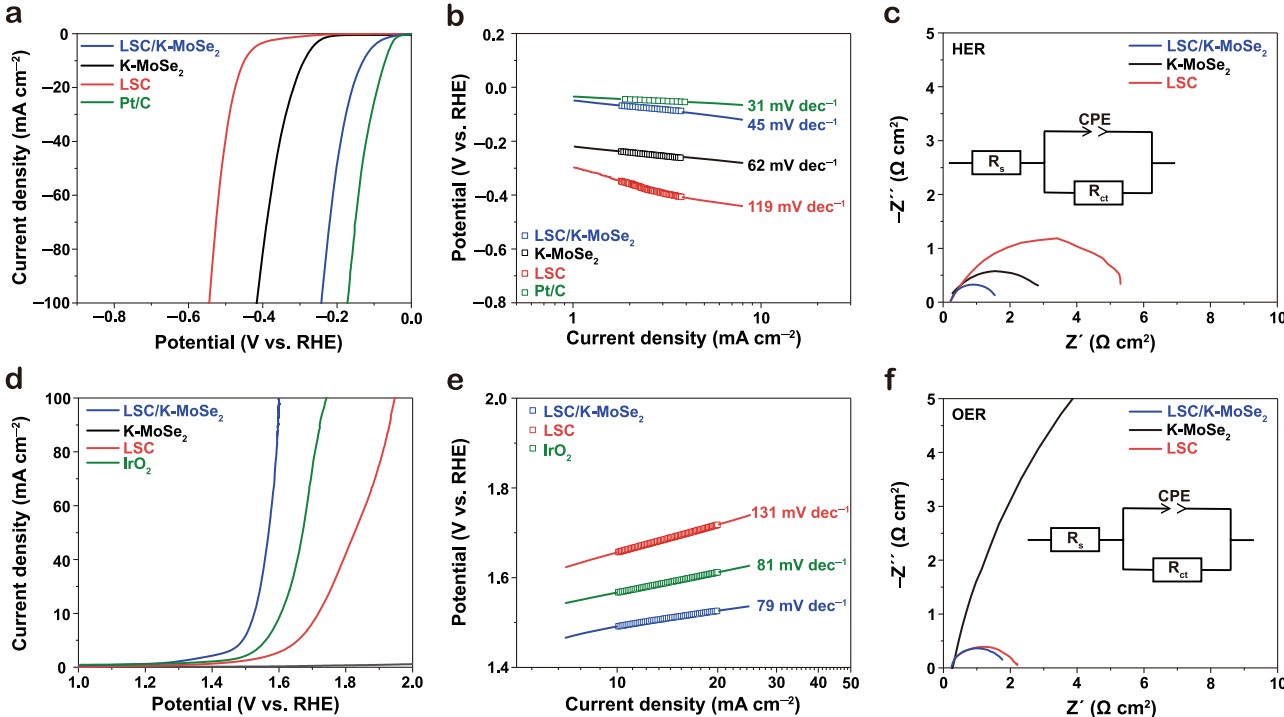

**Fig. 2 Electrocatalytic HER and OER performances of LSC/K-MoSe₂, K-MoSe₂, and LSC. a**, **d** HER and OER polarization curves recorded in N₂-saturated 1 M KOH at a scan rate of 5 mV s⁻¹. **b**, **e** Corresponding Tafel slopes of the HER and OER profiles derived from the polarization curves. **c**, **f** EIS analysis of HER and OER for LSC/K-MoSe₂, K-MoSe₂, and LSC. The Nyquist plots comprise real (Z′) and imaginary (Z″) parts fitted as x and y axes, respectively.

the Gibbs free energy of hydrogen adsorbed on the active site closer to the thermoneutral point (~0 eV). It also enables overpotential reduction for the Volmer reaction owing to the increase in proton concentration of the electronic double layer of the catalyst, thereby improving HER performance[40]. UV–Vis–NIR spectroscopy analysis in Fig. 3f reveals that the A and B excitonic peaks observed for semiconducting 2H-MoSe₂[41] disappear in LSC/K-MoSe₂ and K-MoSe₂, illustrating the metal-like characteristics of the as-synthesized LSC/K-MoSe₂ and K-MoSe₂[42].

X-ray photoelectron spectroscopy (XPS) analysis was performed to further elucidate the origin of the enhanced HER and OER performance of LSC/K-MoSe₂. First, we analyzed the chemical states of Co 2p and O 1s core levels of LSC and LSC/K-MoSe₂ to infer the cause of catalytic electrolysis performance improvement. The Co 2p peaks of LSC and LSC/K-MoSe₂ include Co 2p³⁺, Co 2p²⁺, and two satellite features. As shown in Supplementary Fig. 14, in LSC, Co³⁺ and Co²⁺ peaks are located at 779.6/794.2 and 781.5/797.0 eV, and those of LSC/K-MoSe₂ are located at 780.6/795.2 and 782.5/798.0 eV, respectively, illustrating an upshift of 1 eV in the peak position for the heterostructure. This upshift can be attributed to the electronic interaction between LSC and K-MoSe₂. In LSC/K-MoSe₂, a difference in the electronegativity between Mo and Co induces charge transfer, which can modulate the electronic structure (e_g orbital) of Co while maintaining the overall electroneutrality. As the e_g-orbital filling of perovskite oxide affects the binding of oxygen-related intermediates at the active site (typically at B site), optimizing the e_g-orbital occupancy close to 1 is critical for achieving optimal OER performance[43]. In Co, which is the B site of LSC, e_g-orbital fillings of Co³⁺ ($t^5_{2g}e^1_g$) and Co²⁺ ($t^5_{2g}e^2_g$) are 1 and 2, respectively, which suggests that increasing the Co³⁺ proportion over Co²⁺ is preferable to obtain the optimized e_g-orbital occupancy[44,45]. The ratio of Co³⁺/Co²⁺, the value obtained from the XPS spectra, is summarized in Supplementary Table 5; this ratio was 1.5 and 2.4 for LSC and LSC/K-MoSe₂, respectively.

Therefore, near-unity e_g-orbital occupancy can be expected for LSC/K-MoSe₂, confirming improved OER performance in the heterostructure. The chemical state of O 1s in the catalyst can also directly affect OER kinetics[46,47]. As shown in Supplementary Fig. 15, O 1s peaks of LSC and LSC/K-MoSe₂ include four secondary (shoulder) peaks comprising the lattice oxygen at 528.31 eV (O₂⁻, denoted as LO), highly oxidative oxygen species at 531.1 eV (O₂²⁻/O⁻, denoted as OO), surface adsorbed oxygen including hydroxyl groups at 532.11 eV (O₂/OH⁻, denoted as SO), and adsorbed molecular water at 533.2 eV (H₂O, denoted as AW). The larger amount of surface adsorbed oxygen species compared to the lattice oxygen on the catalyst surface has a favorable effect on the formation of oxygen vacancy and rate-determining step of the OER process[46,47]. Thus, the increased SO/LO ratio of LSC/K-MoSe₂ over LSC indicates that the formation kinetics of the active sites–O, –OH, and –OO bonds for the heterostructure was improved, which can contribute to the enhanced electrolytic performance in alkaline solutions (Supplementary Table 6).

The chemical state of MoSe₂ was also investigated to further elucidate the origin of electrochemical performance improvement in the heterostructure. As shown in Fig. 3g, h, 1T- and 2H-phase MoSe₂ with different relative ratios coexist in LSC/K-MoSe₂ and K-MoSe₂. The XPS peaks of the 1T-phase in these configurations are positioned at 228.3 and 231.4 eV for Mo 3d_{5/2} and Mo 3d_{3/2}, and at 53.7 and 54.7 eV for Se 3d_{5/2} and Se 3d_{3/2}, whereas those of the 2H-phase are located at 229.0 and 232.8 eV for Mo 3d_{5/2} and Mo 3d_{3/2}, and at 54.3 and 55.8 eV for Se 3d_{5/2} and Se 3d_{3/2}, respectively. Figure 3i summarizes the relative contents of the 1T- and 2H-phase MoSe₂ for as-prepared LSC/K-MoSe₂ and K-MoSe₂ obtained from the XPS spectra. The high-purity 1T-phase MoSe₂ (~91%) found in the heterostructure over that of K-MoSe₂ (~67%) clearly evidence that electronic interaction occurs between LSC and K-MoSe₂ causing the further metallic-phase transition of MoSe₂ in K-MoSe₂.

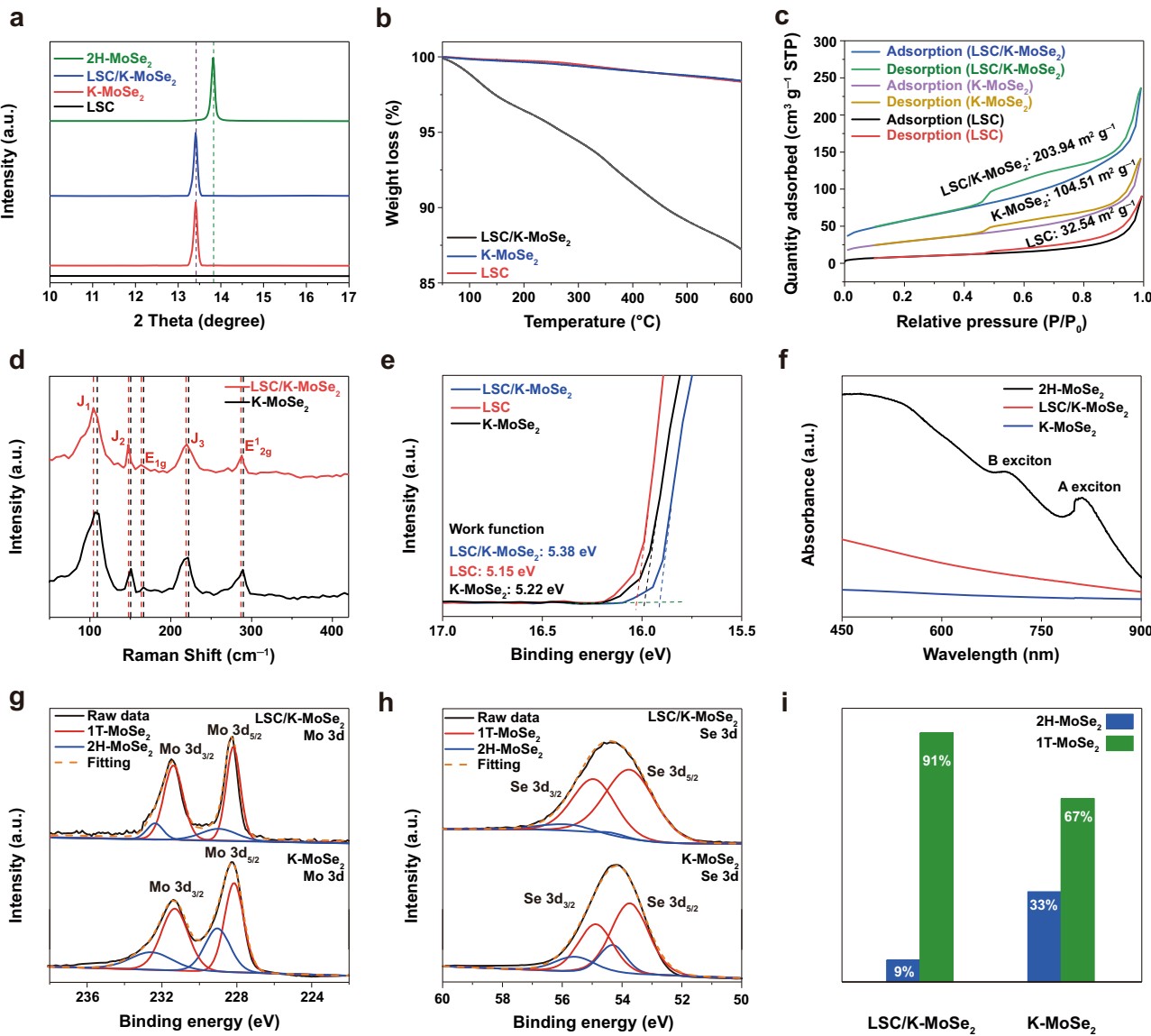

**Fig. 3 Characterization of the LSC/K-MoSe₂ heterostructure. a** XRD patterns for 2H-MoSe₂, LSC/K-MoSe₂, K-MoSe₂, and LSC, demonstrating that each of the K-MoSe₂ and LSC phases are well preserved in LSC/K-MoSe₂. **b** TGA analysis of LSC/K-MoSe₂, K-MoSe₂, and LSC, indicating the improved surface adsorption capability of LSC/K-MoSe₂. **c** BET surface area of LSC/K-MoSe₂, K-MoSe₂, and LSC obtained from N₂ adsorption/desorption isotherms. **d** Raman spectra of LSC/K-MoSe₂ and K-MoSe₂, illustrating the electronic interaction between LSC and K-MoSe₂. **e** UPS valence band spectra of LSC/K-MoSe₂, LSC, and K-MoSe₂. **f** UV–Vis–NIR spectra of 2H-MoSe₂, LSC/K-MoSe₂, and K-MoSe₂, indicating the metallic features of LSC/K-MoSe₂ and K-MoSe₂. **g, h** High-resolution XPS spectra of Mo 3d and Se 3d peaks for LSC/K-MoSe₂ and K-MoSe₂. **i** Relative fraction of 2H- and 1T-MoSe₂ in LSC/K-MoSe₂ and K-MoSe₂, indicating the substantial increase of the 1T-phase ratio in MoSe₂ through complementary charge transfer.

**Complementary charge transfer in LSC/K-MoSe₂.** We hypothesize that the drastic increase in the 1T-phase content of MoSe₂ in the heterostructured catalyst occurs because of the complementary charge transfer between LSC and K-MoSe₂, as schematically illustrated in Fig. 4a. K-MoSe₂ is synthesized by intercalating the potassium metal within the MoSe₂ interlayers, followed by subsequent exfoliation, where potassium atoms and Se form K–Se ionic bonds. The large electronegativity difference (1.73) between K and Se promotes the charge transfer from K to Se, modulating the Mo 4d orbital configuration of MoSe₂ from the occupied $4d_{z^2}$ level to incompletely filled $4d_{xz}$, $4d_{yz}$, and $4d_{yx}$ orbitals. This electronic structure rearrangement causes local phase transition from 2H-MoSe₂ to 1T-MoSe₂. Following the formation of the LSC/K-MoSe₂ heterostructure, charge transfer from the Co of LSC to the 2H-MoSe₂ portion of K-MoSe₂ occurs, causing an additional 1T-phase transition in MoSe₂. This

complementary charge transfer in LSC/K-MoSe₂ has beneficial effects on both HER and OER performance. The increased 1T-phase MoSe₂ concentration in LSC/K-MoSe₂ creates a more electron-rich and electrically conducting surface, enabling fast electron transfer at the catalytically active site, thereby improving HER kinetics[48]. The Co in LSC, after donating the electrons to K-MoSe₂, becomes more electrophilic; such charge transfer results in the upshift of the d-band center and improved adsorption capability of oxygen-generating intermediates (e.g., O*, OH*, OO*) at the catalyst surface[49], thereby enhancing OER kinetics.

We computationally analyzed charge transfer and catalytic activities in LSC/K-MoSe₂ using density functional theory (DFT) calculations. The evaluated charge transfer into a MoSe₂ monolayer from Bader charge analysis[50] is 2.48 $e$ (0.88 $e$ and 1.60 $e$ from the K atom and LSC perovskite, respectively) in the LSC/K-MoSe₂ structure model (Fig. 4b). We found that the

charge transfer from LSC to $MoSe_2$ is unaffected by the K atoms because the LSC to $MoSe_2$ charge transfer in the $LSC/MoSe_2$ structure is the same value of 1.60 $e$, indicating that the complementary charge transfer from K and LSC into $MoSe_2$ is available in the $LSC/K-MoSe_2$ system. The additional electron transfer from K into $MoSe_2$ layers can lead to a larger portion of $1T-MoSe_2$ in $LSC/K-MoSe_2$ than in $LSC/MoSe_2$, supporting the experimentally observed results (Fig. 3i). We found that the charge transfer from LSC to $MoSe_2$ mainly occurs in the $CoO_2$ subsurface layer of LSC as depicted in Fig. 4c, wherein drastic changes in the electron density are observed in the $CoO_2$ layer; these changes are expected to boost OER performance with a change in the electronic structure of LSC layers. Charge transfer from the $CoO_2$ layer to $MoSe_2$ was also confirmed via the density of states (DOS) analysis presented in Supplementary Fig. 16 and Supplementary Note 2. The $CoO_2$ layer under the outermost $La_{0.5}Sr_{0.5}O$ surface plays an important role in charge transfer while electrocatalytic activities occur on the $La_{0.5}Sr_{0.5}O$ surface, which is essential in explaining the superiority of LSC perovskite. Furthermore, we found that the complementary charge transfer from K and LSC into $MoSe_2$ layers markedly reduced the energy barrier when $MoSe_2$ underwent 2H- to 1T-phase transition, as shown in Fig. 4d (0.65 eV in $MoSe_2$ to 0.18 eV in $LSC/K-MoSe_2$); this indicates that a high portion of $1T-MoSe_2$ phase will be present in the $LSC/K-MoSe_2$ heterostructure.

To further elucidate the enhanced water-splitting performance in $LSC/K-MoSe_2$ heterostructure, we analyzed the reaction-free energy for subreactions in both HER and OER. The values of $\Delta G_{H*}$ can directly evaluate HER performance[51–53]. Figure 4e shows a comparison of $\Delta G_{H*}$ in $LSC/K-MoSe_2$ with LSC, 2H-$MoSe_2$, and $K-MoSe_2$. The $LSC/K-MoSe_2$ structures are suitable catalysts for providing the most favorable HER environment for $H_2$ production with the smallest $\Delta G_{H*}$ of 0.25 eV, whereas the $\Delta G_{H*}$ values for LSC, 2H-$MoSe_2$, and $K-MoSe_2$ are 2.01, 1.44, and 0.69 eV, respectively. The optimal near-zero $\Delta G_{H*}$ in $LSC/K-MoSe_2$ is associated with the increased number of states near the energy of normal hydrogen electrode (NHE) potential (Supplementary Fig. 17), thus increasing the interactions between the hydrogen s orbital and the $1T-MoSe_2$ states.

Excellent OER performance in $LSC/K-MoSe_2$ was also explained through atomic-level simulations. Figure 4f–h shows the free energy diagrams for OER in LSC, $LSC/K-MoSe_2$, and $LSC/MoSe_2$, respectively. Our DFT calculations strongly suggest that the OER reactions proceed with the lattice-oxygen participation mechanism[54,55] dominantly rather than the conventional adsorbate evolving mechanism in the $LSC/K-MoSe_2$ system (Supplementary Fig. 19), and the free energy barriers of reaction-determining steps were evaluated to be 2.53, 2.06, and 1.83 eV for LSC, $LSC/MoSe_2$, and $LSC/K-MoSe_2$, respectively. Taking previously reported DFT-calculated OER free energy barriers (2.05 −2.19 eV)[56–58] in $IrO_2$ catalysts into account, the order of computed free energy barriers (LSC > $IrO_2$ > $LSC/K-MoSe_2$) agrees well with the experimental results where the OER onset potentials decrease according to same descending order of LSC > $IrO_2$ > LSC/$K-MoSe_2$, as shown in Fig. 2d. The reduced free energy barrier in the $LSC/K-MoSe_2$ originates from the well-balanced free energies between two governing reactions: (1) OH* to $(V_O + OO)*$ + $H^+$ + $e^-$ and (2) $(H_O + OH)*$ to OH* + $H^+$ + $e^-$, where OH*, $(V_O + OO)*$, and $(H_O + OH)*$ indicate OH absorbed on LSC surface, OO adsorbed on LSC surface with a neighboring oxygen vacancy, and OH adsorbed on LSC surface with a H atom adsorbed on a neighboring lattice O atom, respectively (see Supplementary Fig. 20 for the detailed atomic structures). While the free energy changes in reaction (1) (the red dotted circle) decrease on the order of LSC (2.53 eV) > $LSC/K-MoSe_2$ (1.83 eV) > $LSC/MoSe_2$ (0.67 eV), those in reaction (2) (the violet dotted circle) increase in the opposite

order; LSC (0.99 eV) < $LSC/K-MoSe_2$ (1.39 eV) < $LSC/MoSe_2$ (2.06 eV), effectively producing the lowest free energy barrier in $LSC/K-MoSe_2$ with a remarkably ideal balance between the rate-determining steps of reactions (1) and (2). Reaction (1) involves the replacement of the hydrogen in the OH adsorbate with the lattice oxygen that escapes from the lattice, leaving an oxygen vacancy. Therefore, the free energy barrier in reaction (1) is strongly affected by the energy difference between O* and $(V_O + OO)*$ structures[59] as evaluated in Supplementary Fig. 21, wherein $(Vo+OO)*$ structures are energetically more unfavorable in the same order with the free energy barriers as that in reaction (1); see DOS plots in Supplementary Fig. 22 for additional information on the instability of $(V_O + OO)*$ in LSC. In the case of reaction (2), the free energy barrier is related to how easily a hydrogen atom can be detached from the lattice-oxygen atom; see Supplementary Fig. 23, wherein the energy difference between $(H_O + OH)*$ and OH* increases on the same order of the free energy barriers as that in reaction (2).

**Overall water electrolysis of the $LSC/K-MoSe_2$ couple.** The developed $LSC/K-MoSe_2$ demonstrated excellent intrinsic bifunctional catalytic activity in alkaline electrolytes for both HER and OER. Herein, the overall water electrolysis of the $LSC/K-MoSe_2$ configuration (*i.e.*, $LSC/K-MoSe_2$ used for both cathode and anode, denoted as $LSC/K-MoSe_2||LSC/K-MoSe_2$) was examined to further demonstrate the overall water-splitting performance and stability of $LSC/K-MoSe_2$ as the bifunctional electrocatalyst in $N_2$-saturated 1 M KOH solution. Efficient generation of the hydrogen (cathode) and oxygen (anode) gases using the $LSC/K-MoSe_2||LSC/K-MoSe_2$ couple was confirmed, as illustrated in Fig. 5a and Supplementary Movie 1. Figure 5b shows the cell voltage ($E_{cell} = E_{anode} − E_{cathode}$) measurement results of $Pt/C||IrO_2$ (Pt/C for cathode and $IrO_2$ for the anode) and $LSC/K-MoSe_2||LSC/K-MoSe_2$ obtained during the water electrolysis reaction. Consistent with the result of the half-cell-configured polarization profiles, the $LSC/K-MoSe_2||LSC/K-MoSe_2$ couple demonstrated better overall water electrolysis performance than the state-of-the-art noble-metal-based $Pt/C||IrO_2$ couple. The cell voltages needed to attain 10 and 100 mA cm$^{-2}$ were 1.59 and 1.95 V for $LSC/K-MoSe_2||LSC/K-MoSe_2$ and 1.67 and 2.04 V for $Pt/C||IrO_2$. Although the HER performance of $LSC/K-MoSe_2$ in the half-cell reaction was slightly lower than that of Pt/C, the overwhelmingly high OER performance resulted in excellent overall water electrolysis activity for $LSC/K-MoSe_2$, surpassing that of the noble-metal pair $Pt/C||IrO_2$. In addition to performance, electrochemical stability is an equally important criteria to consider when promoting the broad industrial pertinence of water electrolysis catalysts. To examine the electrochemical stability of $LSC/K-MoSe_2$, its chronopotentiometric profile was measured at a high current density of 100 mA cm$^{-2}$. As shown in Fig. 5c, for the $Pt/C||IrO_2$ reference, a rapid increase in cell voltage, which indicates cell failure, was observed within 60 h under our experimental test conditions. However, the $LSC/K-MoSe_2||LSC/K-MoSe_2$ couple exhibited exceptionally high electrochemical durability even after 2,500 h of continuous operation without noticeable performance degradation. To verify the excellent operational durability of $LSC/K-MoSe_2$ as the electrocatalyst, its physical and chemical characters were investigated after 2,500 h of stability testing. Supplementary Fig. 24 shows the SEM image of the catalyst electrode before and after the stability test. Even after 2,500 h of the electrocatalytic reaction, the starting electrode structure was well preserved without any significant physical damages or detachment. XPS analysis further reveals the superior chemical stability of $LSC/K-MoSe_2$ (Supplementary Fig. 25 and Supplementary Tables 7, 8). After 2,500 h of

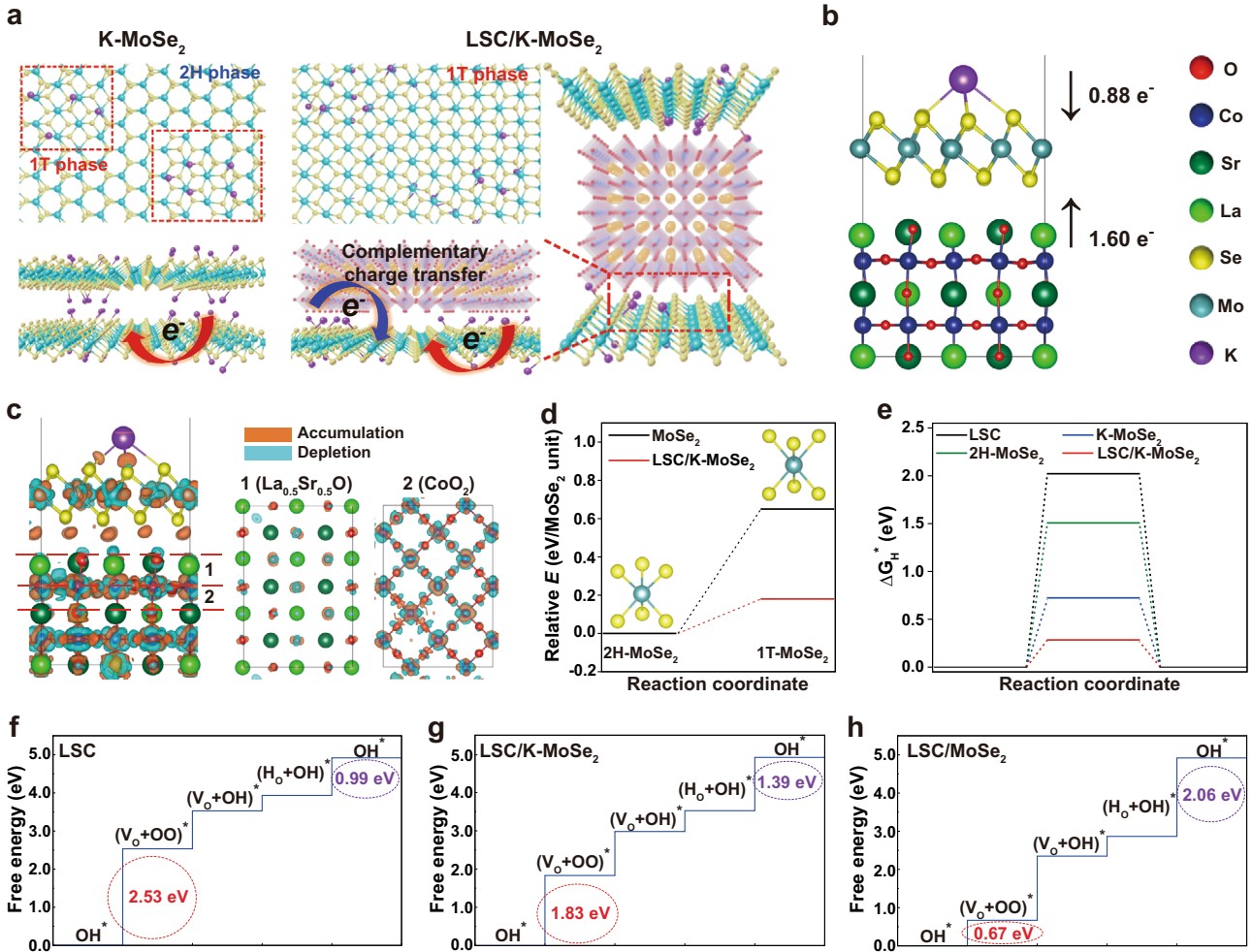

**Fig. 4 Complementary charge transfer phenomena in bifunctional LSC/K-MoSe₂ catalysts. a** Schematic of the atomic structure and charge transfer effect for K-MoSe₂ and LSC/K-MoSe₂. Complementary charge transfer in LSC/K-MoSe₂ can modulate the electronic structure of MoSe₂, increasing the 1T-MoSe₂ ratio in the heterostructure. **b** Charge transfer from K and LSC to MoSe₂ in the optimized LSC/K-MoSe₂ heterostructure. **c** Charge density difference plot for the interface between LSC and K-MoSe₂. Side view of the total structure (left) and the cross section of the layers of La$_{0.5}$Sr$_{0.5}$O (middle) and CoO₂ (right) are demonstrated. **d** Relative energies of 1T-MoSe₂ in monolayer structure (black line) and LSC/K-MoSe₂ heterostructure (red line) compared to the energy of the 2H-MoSe₂ monolayer. **e** Free energy diagrams for HER for LSC, 2H-MoSe₂, K-MoSe₂, and LSC/K-MoSe₂. **f–h** Free energy diagrams for OER for LSC, LSC/K-MoSe₂, and LSC/MoSe₂.

overall water electrolysis reaction, each ratio of the Co$^{3+}$/Co$^{2+}$ in Co 2*p* and surface-active oxygen/lattice oxygen in O 1s exhibited almost negligible changes compared with those of pristine LSC/K-MoSe₂. MoSe₂ may potentially be decomposed into molybdenum oxides and selenate at the potentials of the OER electrode during prolonged electrolysis[60]. To verify the stability of MoSe₂ in LSC/K-MoSe₂, we performed XPS analysis for Mo and Se in LSC/K-MoSe₂ on the OER electrode after 2,500 h of chronopotentiometric stability test. Although some partial oxidization was observed, the integrity of MoSe₂ was well preserved without decomposition (Supplementary Fig. 26 and Supplementary Table 9). Moreover, the energy efficiency of the overall water electrolysis using LSC/K-MoSe₂||LSC/K-MoSe₂ at 100 mA cm$^{-2}$ was calculated to be 75.4% (Supplementary Note 1). Considering the energy efficiency of a typical noble-metal-based water electrolysis catalyst is around 70%, our results demonstrate that the developed bifunctional LSC/K-MoSe₂ catalyst can be used as a promising electrocatalyst in water electrolysis for efficient hydrogen production. Figure 5d (also summarized in Supplementary Table 10) compares the electrochemical stability of overall water electrolysis for various catalyst configurations

reported to date along with the proposed LSC/K-MoSe₂ in this work. Additionally, we further investigated the operational durability of the LSC/K-MoSe₂ in various harsh environments. Chronopotentiometric stability test of the LSC/K-MoSe₂ couple was conducted under high operational temperature and current density (500 and 1,000 mA cm$^{-2}$ in 1 M KOH at 60 °C) and high electrolyte concentration (100 mA cm$^{-2}$ in 10 M KOH at room temperature). In 1 M KOH at 60 °C, the LSC/K-MoSe₂ couple required cell voltages of 2.25 V at 500 mA cm$^{-2}$ and 2.52 V at 1000 mA cm$^{-2}$, respectively (Supplementary Fig. 27). Under such conditions, it exhibited stable operational stability over 1,200 and 800 h at 500 and 1,000 mA cm$^{-2}$, respectively, without obvious performance degradation (Supplementary Fig. 28). In 10 M KOH at room temperature, the cell voltage needed to achieve 100 mA cm$^{-2}$ was 1.87 V for the LSC/K-MoSe₂ couple (Supplementary Fig. 29). In the two-electrode cell, the chronopotentiometric stability of 1,600 h was achieved at 100 mA cm$^{-2}$ in 10 M KOH at room temperature without any noticeable performance degradation (Supplementary Fig. 30). Despite the various accelerated test conditions, our heterostructure demonstrates overwhelmingly superior durability for water electrolysis.

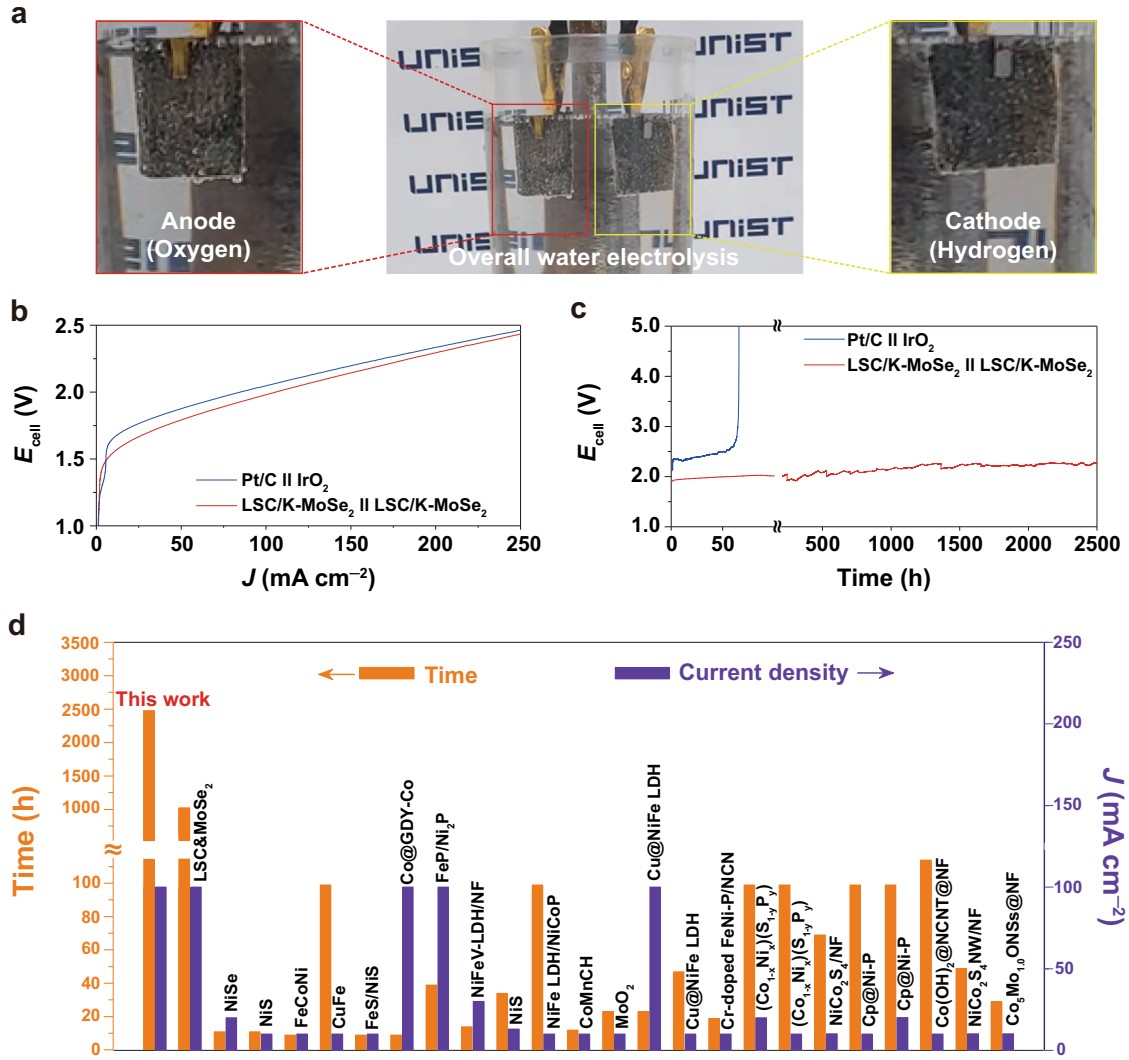

**Fig. 5 Overall water electrolysis of LSC/K-MoSe₂. a** Digital image of full-cell water-splitting system comprising a two-electrode configuration with LSC/K-MoSe₂||LSC/K-MoSe₂. **b, c** LSV and chronopotentiometric durability curves of Pt/C||IrO₂ and LSC/K-MoSe₂||LSC/K-MoSe₂ measured in 1 M KOH. **d** Summary of the overall water electrolysis stability of LSC/K-MoSe₂||LSC/K-MoSe₂ and other reported electrocatalyst couples.

## Discussion

In this work, we developed a heterostructure-based electrocatalyst that demonstrates excellent overall water electrolysis performance and stability. The complementary charge transfer induced within the heterogeneous catalyst, comprising alkali-metal-treated transition-metal dichalcogenides and perovskite oxide, generated synergistic effects in both HER and OER processes. The modulated electronic structure of MoSe₂ with high metallic-phase purity and improved electrical conductivity enhanced HER kinetics. The increased electrophilicity of LSC improved the adsorption capability of oxygen-generating intermediates on the catalyst surface, thereby boosting OER performance over that of IrO₂. The water electrolysis performance using the LSC/K-MoSe₂||LSC/K-MoSe₂ couple outperformed the state-of-the-art noble-metal pair of Pt/C||IrO₂, exhibiting lower cell voltage for the overpotential at 10 and 100 mA cm⁻², improved energy efficiency, and excellent operational stability over 2,500 h. This work can provide a promising perspective for the performance maximization of heterostructure-based catalysts in water electrolysis to substitute precious-metal-based electrocatalysts.

## Methods

**Synthesis of K-MoSe₂.** For the preparation of K-MoSe₂, 500 mg of bulk MoSe₂ (<2 μm, purity >99%, Alfa Aesar) and 200 mg of potassium metal (stored in oil, purity >99.95%, Kojundo, Korea) were added to a glass tube under inert conditions in a glovebox. The tube was sealed and treated at 400 °C for 1 h. The as-prepared potassium-intercalated MoSe₂ was rinsed with deionized water and ethanol to remove potassium ion residues. The resulting powder was dried at 80 °C for 24 h.

**Synthesis of LSC.** To synthesize the LSC perovskite oxides, an aqueous solution containing dissolved La, Sr, and Co nitrites (La(NO₃)₃·6H₂O, Sr(NO₃)₂, Co(NO₃)₂·6H₂O, Alfa Aesar) in stoichiometric amounts and citric acid (C₆H₈O₇, Sigma-Aldrich) in deionized water was prepared. Post solvent evaporation, the resulting wet-gel was calcined at 900 °C for 2 h and at 950 °C for 10 h to remove the organic fraction. Finally, the resulting reaction product was mortared to homogenize the LSC.

**Synthesis of LSC/K-MoSe₂.** To synthesize the LSC/K-MoSe₂ heterostructure, as-prepared LSC and K-MoSe₂ were high-energy milled with 10 wt.% of Ketjen Black (KB) (EC-600JD, Lion Specialty Chemicals Co. Ltd.) via the planetary ball mill system (PM-100, Retsch). To find the optimized weight ratio for LSC and K-MoSe₂ for electrochemical performance, various mixture weight ratios were examined with LSC:K-MoSe₂:KB of 8:1:1, 7:2:1, 6:3:1, and 5:4:1. The total weight of the mixture was maintained as 600 mg. LSC, K-MoSe₂, and KB in ethanol were sealed into the steel jar and ball-milled at 500 rpm for 2 h. After completing the ball-milling process, the resulting product was thoroughly dried and collected.

**Material characterizations**. The morphology and EDS elemental mapping of the catalysts were characterized using HR-TEM (JEM-2100F, JEOL) with an accelerating voltage of 200 kV. Crystallographic information of catalysts was analyzed through high-power XRD (D/MAX2500V/PC, Rigaku) at 40 kV and 200 mA at a scanning rate of 1° min$^{-1}$ in the diffraction range of 10°–80°. The chemical state and work function were investigated via XPS (ESCALAB 250XI, Thermo Fisher Scientific) with monochromated Al-Kα radiation. TGA analysis was conducted to investigate the surface adsorption capacity of catalysts at a ramping temperature rate of 10 °C min$^{-1}$ using thermo-gravimetric analyzer (Q500, TA). The surface area and pore size of the catalysts were evaluated using $N_2$ desorption/adsorption isotherms of catalysts using a physisorption analyzer (ASAP 2420, Micromeritics Instruments). The optical properties of the catalysts were collected using a UV–Vis–NIR spectrophotometer (Cary 5000, Agilent). Raman spectra were recorded using confocal Raman spectroscopy (Alpha300R, WITec) equipped with a 532-nm laser. The electrode morphologies before and after overall water electrolysis were obtained via cold FE-SEM (S-4800, HITACHI).

**Electrochemical measurements**. Half-cell electrochemical measurements were performed in 1 M KOH, in which saturated Ag/AgCl and a carbon rod were used as the reference and counter electrodes, respectively, in a three-electrode configuration controlled by an electrochemical workstation (CHI 760E, CH Instruments Inc.). Catalyst ink including LSC/K-MoSe$_2$, K-MoSe$_2$, and LSC was prepared by dispersing 9 mg of the catalyst and 1 mg of KB in a 1 mL binder solution comprising 5 wt.% Nafion solution (Sigma-Aldrich), ethanol, and isopropyl alcohol, followed by bath sonication. A catalyst ink of Pt/C and IrO$_2$ was similarly prepared except for the KB. The working electrode was prepared via drop-casting 5 μL of the as-prepared catalyst inks onto the glassy carbon disk electrode with an area of 0.071 cm$^2$. The HER and OER LSV polarization curves were obtained at a scan rate of 5 mV s$^{-1}$ in N$_2$-saturated electrolyte, which were measured from 0 to −1.0 V (vs. reversible hydrogen electrode (RHE)) and from 1.0 to 2.0 V (vs. RHE), respectively. Tafel slope values were derived from the LSV curves by plotting the overpotential against current density in log-scale from 1 to 10 mA cm$^{-2}$. All potentials in this work were measured with respect to Ag/AgCl reference electrode and converted to RHE scale using the following formula in 1 M KOH (pH 14): $E$(vs. RHE) = $E$(vs. Ag/AgCl) + $E_{Ag/AgCl}$ (=0.197 V) + 0.0592 pH = $E$(vs. Ag/AgCl) + 1.0258 V. EIS measurements were conducted at an overpotential of −0.2 and 0.7 V (vs. RHE) for HER and OER, respectively, in a frequency range of 100 kHz to 0.01 Hz with an amplitude of 10 mV in 1 M KOH. All half-cell polarization curves were corrected for ohmic losses by the following equation. $E = E$(RHE) – $iR_s$, where $E$ is the potential after the $iR$-correction, $E$(RHE) is the measured potential with respect to RHE (before $iR$-correction), $i$ is the measured current, and $R_s$ is the uncompensated resistance obtained from EIS analysis. To measure the double-layer capacitance ($C_{dl}$) value, the potential window of cyclic voltammograms was cycled in the non-Faradaic region from 0.03 to 0.33 V (vs. RHE) with different scan rates from 20 to 160 mV s$^{-1}$. $C_{dl}$ values were derived by plotting the charging current density difference ($\Delta j = (j_a - j_c)/2$) at 0.18 V. Overall water electrolysis testing was conducted using a two-electrode configuration comprising electrosprayed catalyst inks on Ni foam (with a catalyst loading of 1 mg cm$^{-2}$) as the current collector. Chronopotentiometry stability test results of the overall water electrolysis were obtained under a current density of 100 mA cm$^{-2}$ using an electrochemical workstation (ZIVE BP2C, Wonatech Co., Ltd.).

**Computational details**. We performed spin-polarized ab initio calculations using the Vienna ab initio simulation package (VASP)[61] within the projector augmented wave (PAW) method[62] and Perdew–Burke–Ernzerhof (PBE)[63] exchange and correlation functionals. The DFT+$U$ method based on Dudarev's approach[64] was adopted with $U$ = 4.3 and $J$ = 1.0 eV ($U_{eff}$ = 3.3 eV) for Co-3$d$ and $U_{eff}$ = 4.0 eV for Mo-4$d$, as employed in previous studies[65,66]. First, the LSC slab structure was prepared based on 2√2 × 3√2 × 2 supercells with >20 Å vacuum space. A 1 × 1 × 1 Monkhorst-pack $k$-points mesh was adopted with 2 × 10$^{-2}$ eV Å$^{-1}$ for force criterion in the ionic relaxations and a 400 eV energy cutoff for the plane-wave basis set was used with valance electron configurations of 5$s^2$5$p^6$5$d^1$6$s^2$ (La), 4$s^2$4$p^6$5$s^2$ (Sr_sv), 3$d^8$4$s^1$ (Co), 2$s^2$2$p^4$ (O), 4$s^2$4$p^6$4$d^5$5$s^1$ (Mo_sv), 4$s^2$4$p^4$ (Se), and 3$s^2$3$p^6$4$s^1$ (K_sv) orbitals for La, Sr, Co, O, Mo, Se, and K, respectively. The LSC/MoSe$_2$ and LSC/K-MoSe$_2$ structures were modeled using 2 × 5 supercell of 2H- or 1T-MoSe$_2$ monolayer placed on the LSC (001) surfaces with <3% lattice mismatch for LSC/K-MoSe$_2$ (the length of the cell-vectors: $a$ = 11.20 Å, $b$ = 16.57 Å, and $c$ = 35.00 Å), K atoms were attached on the MoSe$_2$ monolayer at the energetically most stable site, where one K atom was adsorbed on the 2 × 5 MoSe$_2$ supercell surface (Fig. 4b). The top two atomic layers of LSC slab structure were relaxed to simulate the heterostructures. The charge transfer in LSC/K-MoSe$_2$ structure was analyzed using Bader population analysis[50]. Second, the free energy of HER reactions was computed until the residual force components were within 5 × 10$^{-3}$ eV Å$^{-1}$ using the equation $G_H = E(H) + 0.24$ eV, wherein $E(H)$ is the adsorption energy of a H atom, calculated for 1/2H$_2$ at pH = 0 and $p$(H$_2$) = 1 bar, and 0.24 eV correction is for the differences in zero-point-energy and entropy[67,68]. The free energy of each sub-reaction for OER was evaluated using the equation $G = \Delta E + \Delta ZPE - T\Delta S$ at pH = 0, $T$ = 298 K, and zero applied potential (0 V vs. RHE), where $\Delta G$, $\Delta E$, $\Delta ZPE$, $T$, and $\Delta S$ represent the change in free energy, total energy difference, change in zero-point-energy, temperature, and change in entropy, respectively. The $\Delta ZPE$ and $T\Delta S$ terms were adopted from previous studies and gas phase H$_2$O and O$_2$ were considered references for all the reactions, assuming 0.035 bar H$_2$O gas pressure at room temperature for equilibrium with liquid water. The free energy O$_2$

gas was computed by fixing the free energy changes in the overall reaction (H$_2$O → 1/2O$_2$ + H$_2$) to the experimentally measured value, 2.46 eV, as adopted in previously reported OER computations[59,69–73]. To model OER on LSC (001) surfaces in LSC/K-MoSe$_2$, we used the cropped 2 × 1.5 MoSe$_2$ supercells to expose the LSC (001) surface to secure enough space for OER adsorbates on the LSC surface. A 4 × 4 × 1 Monkhorst-pack $k$-points mesh was used to analyze the DOS.

## Data availability
The data measured, simulated, and analyzed in this study are available from the corresponding author on reasonable request.

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

## Acknowledgements

This work was supported by Basic Science Research Program through the National Research Foundation of Korea (NRF) grant funded by the Korea government (MSIT) (No. 2019R1A2C1009025, 2019R1A4A1029237, and 2020R1A4A3079710). Y.-K.H would like to thank Bohyun Hwang and Sujin Lee for their technical support.

## Author contributions

N.K.O. carried out most of the experimental work and wrote the manuscript. J.S. wrote the manuscript and contributed to the designing of schematics. S.L. and H.-J.K. performed the DFT calculation and wrote the manuscript. U.K. helped the interpretation of electrochemical analysis. J.L. performed XPS measurements. Y.-K.H. directed the theoretical work. H.P. conceived the project and directed the overall work and manuscript writing. All the authors contributed to the discussion and analysis of the results regarding the manuscript.

## Competing interests

The authors declare no competing interests.
