## [Peer Review File · Nature Communications]

REVIEWER COMMENTS

Reviewer #1 (Remarks to the Author):

Firstly, this article bears close similarity to the results published by the same group of authors in 2019.

Oh, Nam Khen, et al. "In-situ local phase-transitioned MoSe₂ in La_{0.5}Sr_{0.5}CoO_{3-δ} heterostructure and stable overall water electrolysis over 1000 hours." *Nature communications* 10.1 (2019): 1-12.

It is not obvious to the reviewer as to why such a closely related paper from their own work has not even been referenced or discussed. I find this oversight surprising.

It is inaccurate to state that Pt and IrO₂ are unstable for water electrolysis. These catalysts are quite stable under the conditions of an acidic PEM electrolyzer. Please see the work of Proton Onsite. Proton has shown durability at 2400 psi differential pressure to over 20,000 hours with these catalysts. Please see "Research Advances towards Low Cost, High Efficiency PEM Electrolysis" Kathy Ayers et al, 2010 *ECS Trans.* 33 3.

MoSe₂ is a well known catalyst for HER but is not stable under OER conditions. MoSe₂ will be oxidized or transformed to soluble molybdate and selenate at the potentials of the OER electrode during prolonged electrolysis. Analyzing the solution for Mo(VI) species after extended electrolysis would be necessary. XPS data on the electrodes following electrolysis studies for Mo and Se are not provided.

Under these circumstances, any enhancement in activity of the oxygen evolution electrode cannot be attributed to the heterostructure-based electronic effects of MoSe₂ material that is actually chemically unstable under the operating conditions of the oxygen electrode.

Since LSC is relatively stable under the conditions of oxygen evolution it is likely that all the activity for OER comes from LSC, and the observed improvement in activity is associated most likely with the increased electrochemically active surface area that results after the leaching of the MoSe₂. Dopant effects of molybdenum could also be beneficial to improving the activity of LSC.

Reviewer #2 (Remarks to the Author):

In the submitted manuscript, the authors reported a heterostructured catalyst of perovskite oxides and MoSe₂, and demonstrated its high activity and stability for catalyzing both oxygen evolution reaction (OER) and hydrogen evolution reaction (HER) for overall water splitting. They attributed the superior performance to the charge transfer between the two compounds. While the catalyst performance appears outstanding among literature and will be of great interest to the researcher in this field, I have several questions or comments on the experiment design and data interpretation.

1. From the perspective of materials chemistry, MoSe₂ as a dichalcogenide, while it may be stable in HER conditions, tends to be less thermodynamically robust than oxides materials in OER conditions. Therefore, questions regarding the true active materials are not uncommon for water-splitting bi-functional catalysts and this usually requires surface-sensitive characterization of the catalysts before and after the test. (see *ACS Energy Lett.* 2017, 2, 8, 1937–1938)

The authors indeed measured the XPS data of Co and O and characterized its morphology by SEM after 2500-hour testing, concluding "high electrochemical durability". However, questions about the possible oxidation of MoSe₂ under OER conditions remain to be addressed. In other words, is the MoSe₂ in the as-synthesized composite really the active catalyst? Is it possible that molybdenum oxide plays a role here?

2. Surface areas:

A. In Figure 3c, LSC/K-MoSe₂ shows larger BET surface areas than the individual constituents. Could this result from the ball-milling process? A control experiment where ball-milled LSC and K-MoSe₂ are prepared and tested is needed to rule out the effect of surface area on performance before considering any electronic interaction effect.

B. In Supplementary Figure 6, "... from which the electrochemically active surface area is estimated.", where are the data? How do those surface areas compare to BET surface area? Which surface area is used in the current density calculation for all electrochemical data, geometric area of the electrode, electrochemically active surface area, or BET surface area?

3. On page 9, the authors considered the change in work functions as a strong indicator of charge transfer. However, only the work functions of LSC and LSC/K-MoSe₂ are given. What is the value for K-MoSe₂? Does it agree with the calculation result that electron transfers from LSC to K-MoSe₂?

4. On the top of Page 20, the iR compensation method is not correct. The given formula is for converting potential to the RHE scale.

5. In Supplementary Figure 10, why the baseline on the left XPS data used linear while the one on the right did not? The Co 2p data in Supplementary Figure 20 also used a linear baseline. If its baseline is changed to linear, how would the distribution of Co oxidation state change in Supplementary Table 3?

6. In Supplementary Figure 14, "All calculations were performed at pH = 0, T = 298 K, and zero applied potential (0 V vs RHE)", is it possible to calculate the free energy diagrams at experimental conditions (pH14, >1.23 V vs RHE)?

Reviewer #3 (Remarks to the Author):

The manuscript "Multidirectional Charge Transfer in Perovskite Oxide and Transition Metal Dichalcogenides Heterostructure for Enhanced Water Electrolysis with Long-Term Operational Stability" deals with the hydrogen evolution (HER) and oxygen evolution (OER) performance of a combined LSC/K-MoSe₂ catalyst. The authors find in their experiments that the catalyst exhibits satisfactory HER performance, compared to Pt, and very good OER performance compared to IrO₂. The authors also show that their studied catalyst exhibits long-term stability under reaction conditions. They then compare these experimental findings with DFT calculations, both to explain the general trends for the catalyst surface, but also to elucidate the pathways present for the HER and OER reactions.

The results shown should be of wider interest in the scientific community due to their link to hydrogen generation and renewable energy utilization.

In principle I think this manuscript presents interesting findings from an experimental viewpoint, however the computational side, in my opinion, does not back up these findings satisfactorily in the current state.

My main issues in regards to the theory part are the choice of the water reference (gas-phase water as a product is not adequate for electrochemical modelling) as well as the unreflected acceptance of the O₂-species present in the calculations, in light of the well-known overbinding effects of O₂ species. Lastly, the calculated onset potentials for the OER, as shown in the SI are neither discussed, nor are they mirroring the, very well performing, experimental catalyst. This suggests to me that the reaction pathway studied in the DFT does not completely match the actual active site or reaction pathway.

The changes necessary to address these issues would be extensive, therefore I recommend major

revisions to the manuscript.

Below please find a more detailed list of comments:

- 1) It would be good to cut the abstract slightly, since it seems a bit long (or at least exactly on the length-limit) at this point. E.g. the first two sentences can be combined, since they are merely expository
- 2) It might be worthwhile to discuss the DFT-calculated OER performance of the IrO₂ reference catalyst and, later on, how that performance differs from the current calculation results.
- 3) "Since the HER performance of 141 LSC/K-MoSe₂ approaches close to that of Pt/C, ...", I find this statement a bit too optimistic, given the numbers shown before. However, I do agree that the results are interesting when comparing to the two constituent elements.
- 4) Purely for reference: the °C sign did not make the conversion into the PDF file (p. 8 – line 180), this also applies to all other °C occurrences later in the text.
- 5) "The mesoporous-type electrocatalyst can effectively improve the surface area of the catalyst as well as the diffusion of the electrolyte and ionic species²⁹" (page 9, line 190/191) – please rephrase this sentence to fix the grammar (e.g. "The mesoporous nature of the catalyst can effectively ...")
- 6) "increasing the exchange current density and enabling the Gibbs free energy of H* (* = active site) closer" – in this case it might be more readable to simply state that this is "the Gibbs Free Energy of hydrogen adsorbed on the active site" since H* is never used again. (Page 9, l. 211)
- 7) It might be good to mention in 1-2 words, which method was used for the charge calculation in the text (p. 12, l. 280)
- 8) In this regard, I had one question: The authors use Bader analysis to assign charges in the surface region of the model catalyst. Bader analysis, as implemented in VASP, in some cases assigns charges from the vacuum region to specific surface atoms. Did the authors make sure to check the volumes assigned to each of the atoms to make sure that they remain constant or at least comparable across the systems they compare?
- 9) "The CoO₂ subsurface plays an important" (p. 13, l. 293) – would it make more sense to call this the CoO₂ substrate?
- 10) In the lines preceding this, the authors state " Charge transfer from the CoO₂ layer to MoSe₂ was also confirmed via the density of states (DOS) analysis presented in Supplementary Figure 12." – Could the authors please also reference the Supplementary Note 2 in which the charge transfer mechanism is explained? With the current information it is very hard to make this connection. Also, please add a note referencing the Supplementary Note 2 in the caption of Supplementary Figure 12. Also, it might be very helpful to turn the discussion in Supplementary Note 2 into a short cartoon pointing out the location of the transferred charges in a direct comparison of the pDOS's, but that would be optional.
- 11) "The optimal near-zero ΔGH" (p. 13, line 306) – I find this, again, a bit too optimistic to call a value of 0.25eV "near-zero". This becomes especially problematic when considering that electrochemical DFT values are usually assumed to have an error bar of ~0.2eV (at worst) and can therefore be as high as ~0.3-0.4eV.
- 12) At the end of page 13 the authors state that "Our DFT calculations strongly suggest that the OER reactions proceed with the lattice-oxygen participation mechanism^{49,50} dominantly rather than the conventional adsorbate evolving mechanism in the LSC/K-MoSe₂ system (see Supplementary Figure 14), and the free energy barriers of reaction-determining steps were evaluated to be 2.53, 2.06, and 1.83 eV for LSC, LSC/MoSe₂, and LSC/K-MoSe₂, respectively, which agrees well with the experimental results that LSC/K-MoSe₂ provides much higher OER performances than LSC and LSC/MoSe₂." This statement is one of my main issues with the manuscript at this point: The authors state that experimentally, the onset potential for the OER is found to be 1.44V (1.21V + 230mV overpotential, as per line 154). However, the DFT finds the onset potential to be around 1.8V. However, even assuming the error bars discussed before, this value does not correspond to the experimental findings and indicates some issue with the pathway as it is shown here.

The discrepancy can have a multitude of causes, however I would point out a few problems with the

current DFT calculations that come to mind and that should be addressed going forward:

12.1) The authors, at this point, do not consider solvation corrections, from what I can see (unless they are included in the ZPE or TS corrections, which would be odd). This will shift the relative energies of OH and OOH the most and will significantly impact the onset potential, which is almost always determined by the step $\text{OH}^* \rightarrow \text{O}^*$

12.2) The authors use gas-phase water as a reference for their calculations. This makes sense for thermocatalysis, however electrocatalytic reactions should be referenced to liquid water.

12.3) The authors state in the methods section that they reference to O_2 in the gas-phase. I assume the authors are aware of the issues with calculating O_2 as a reference in gas-state and are using any of the well-tested methods to avoid this issue. If so, please state this and/or reference to the relevant literature. Otherwise, please address this issue.

12.4) The authors calculate an O_2 species bound to the surface as an intermediate. In this case the same problems as for calculating $\text{O}_2(\text{g})$ will arise unless it is not a neutral O_2 species but e.g. an $\text{O}_2(-)$ species. This can be extracted from e.g. Bader analysis or even better by measuring the O-O distance. If the O-O distance is similar or equal to $\text{O}_2(\text{g})$ then the O_2 intermediate will probably experience overbinding effects, which would then make it appear too stable. If this is the case then the 1.83V onset potential would change drastically.

12.5) The authors combine the two fundamental reaction steps of the step ($\text{O}_2^* \rightarrow \text{OH}^*$) – bottom left to bottom right of SI figure 15. The two steps should be ($\text{O}_2^* \rightarrow \text{O}_{\text{vac}} + \text{O}_2(\text{g})$) and ($\text{O}_{\text{vac}} + \text{H}_2\text{O} \rightarrow \text{OH}^* + \text{H}^+ + \text{e}^-$). The reason why it would be important to separate these steps is because the first one is a chemical desorption step, which is potential independent and the second one is an electrochemical step that is potential dependent. Combining the two can make it appear as if the step is more favorable than it really is once potential is applied.

Once these issues are addressed, I would ask the authors to add several sentences discussing the (new) results for the onset potentials of the OER and how this compares in an absolute scale to the experimental results for the onset potential and where the possible, remaining, differences could lie .

13) The authors state that they use a U value of 4.3eV and a J value of 1.0eV. How were these values determined? Is there a reference for this?

14) It is always a good idea to provide the length of the cell-vectors in Angstrom, since allows the reader to quickly assess if the k-point sampling used is adequate.

15) It might be more helpful to the reader to directly name the POTCAR file that was used for each element (e.g. La_sv), however this can also be done in the SI.

16) I want to point out that currently Figures 2,3 and 4 and the TOC figure exhibit a slightly yellow background for some reason. This is not the case for Figures 1 and 5. This also applies to some of the SI figures as well.

17) All structures files for the intermediates need to be made available, either as an attached zip file or as POSCARs added to the end of the supplementary information.

Apart from these issues, which should be fixable, the manuscript was well-written and organized and I enjoyed reading it.

REVIEWERS' COMMENTS

Reviewer #1 (Remarks to the Author):

Reviewer 1- Response.

I appreciate the authors' detailed response to comments. Listed below are other concerns that need to be addressed before publication.

Response to Comment 1. It is important that the five differences that the authors state between the earlier paper (Nature communications 10.1 (2019): 1-12.) and current work be summarized in the introduction section of the manuscript. Otherwise, the current manuscript seems quite similar to the previous one.

Response to Comment 2. OK.

Response to Comment 3.

The XPS data is quite convincing that there is still molybdenum selenide after 2500 hours of testing. However, it would be helpful to show the XPS data for before and after electrolysis in the same chart.

Only oxides and oxidized species are possible under OER conditions. Reduced state of Mo and Se are not tenable at these potentials. In the added text, please modify the statement "MoSe₂ may potentially be decomposed into Mo and Se at the potentials of the OER electrode ..." to "MoSe₂ may potentially be decomposed into molybdenum oxides and selenate at the potentials of the OER electrode"

Response to Comment 4. The enhancement of the electrode kinetics by molybdenum is central to the conclusions in this manuscript. The speculation about how the Mo increases the OH and OOH adsorption is quite plausible, as the rate-determining step involves the oxidation of adsorbed intermediates. However, the reviewer cannot find any experimental basis for this proposed mechanism. For example, have the authors found differences in Tafel Slope, lowering of adsorption energy of the surface intermediates, or the shifts in binding energy of cobalt in XPS that support this hypothesis? It would be important to provide some discussion on this in addition to speculating the mechanism. For guidance, please see: Malkhandi, S., et al. "Design insights for tuning the electrocatalytic activity of perovskite oxides for the oxygen evolution reaction." The Journal of Physical Chemistry C 119.15 (2015): 8004-8013.

Reviewer #2 (Remarks to the Author):

I'm glad to see all the efforts the authors have put in to improve the manuscript. They have addressed all my comments and I recommend its publication in Nature Comm.

Reviewer #3 (Remarks to the Author):

Reviewer #3:

First of all I want to thank the authors for taking the time to discuss and address the comments made by all reviewers in detail.

I think the changes made to the manuscript significantly strengthened it and I would at this point fully support a publication in Nat Comm.

I had some minor comments left in regards to some of the answers the authors gave, which I will add below. I will leave it to the discretion of the authors on what level they address these.

===

2. It might be worthwhile to discuss the DFT-calculated OER performance of the IrO₂ reference catalyst and, later on, how that performance differs from the current calculation results.

We thank the reviewer for the constructive comment. Previously reported free energy barriers in OER on IrO₂ surfaces are in the range of 2.05–2.19 eV^{R21-R23} at zero applied potential from DFT calculations, which is somewhat higher than the experimentally reported onset potentials, 1.51–1.58 V^{R22,R24-R26}. The most noticeable difference in OER process between IrO₂ and our system (LSC/KMoSe₂) is their oxygen generation pathways; while lattice-oxygen participation mechanism (LOM) prevails in LSC/K-MoSe₂, the conventional adsorbate evolving mechanism (AEM) is more favorable for IrO₂. The DFT-calculated free energy barrier for LOM is much higher (3.63 eV) than that (2.11 eV) for AEM^{R21}. In fact, the order of OER free energy barriers from our computations and the previous DFT-calculated reports (LSC > IrO₂ > LSC/K-MoSe₂) is well agreed with our experiment results (Figure 2d). We added a sentence explaining this point in the revised version of manuscript.

For me personally, at this point, the best reported onset potentials for the OER on IrO₂ would be around 1.53V by Goddard and coworkers (JACS 2016) by considering an OO⁻-type structure, not completely dissimilar to the one discussed here for the studied catalyst. Similar structures were also reported for RuO₂ by Rossmeisl(PCCP, 2014, DFT only) and then later Rossmeisl and Shao-Horn (EES 2017, DFT&Experiment) on RuO₂.

Now it can be argued if the results shown there are really convincing, however, there are at least some indications that DFT and experiment for these systems can be brought to overlap more closely.

It might be interesting to discuss the current results in relation to these results and what trends could be discerned from them in relation to the current results, but I would see this as optional.

11. "The optimal near-zero ΔGH^* " (p. 13, line 306) – I find this, again, a bit too optimistic to call a value of 0.25eV "near-zero". This becomes especially problematic when considering that electrochemical DFT values are usually assumed to have an error bar of ~ 0.2 eV (at worst) and can therefore be as high as ~ 0.3 - 0.4 eV.

We appreciate the reviewer's constructive comment. We agree with the reviewer's point that the $\Delta GH^* = 0.25$ eV may be quite far from "near-zero". While revising the sentence, we updated the ΔGH^* of LSC/K-MoSe₂ with more rigid ion relaxation criterion (force convergence < 0.005 eV/Å) than that in the original manuscript. The newly computed ΔGH^* of LSC/K-MoSe₂ was 0.11 eV, undoubtedly supporting its outstanding experimental HER performance comparable to Pt/C catalysts. It is also found that the decreased force convergence criterion shows the H adsorption status more clearly, leading to the decreased ΔGH^* in LSC/K-MoSe₂ where H atoms have negative adsorption energy ($\Delta E_{H^*} = -0.13$ eV), and the increased ΔGH^* in LSC, 2H-MoSe₂, and K-MoSe₂ where H atoms cannot be stably adsorbed ($\Delta E_{H^*} = 2.25, 1.85,$ and 0.98 eV, respectively). We updated the newly computed ΔGH^* value and revised the sentences; we deleted "near-zero" to minimize any potential confusion. We also updated the computational details and added all the relaxed structures in Supplementary Information.

While I strongly appreciate the authors re-relaxing the structure for this intermediate with a stricter criterion and the following reduction in the ΔGH^* to 0.11eV, I want to ask them consider to move this discussion purely to the SI and leave the initial value in the main text.

The reason for this request is simply that the chosen relaxation criterion in this single case makes the results not 100% comparable to all the other results in the study, which can lead to inconsistencies. However, I find the result very positive in relation to the initial consideration in my question and I thank the authors for taking the time to recalculate this.

Comment 12:

I thank the authors for further specifying their methods section to reflect the true extent of effort and expertise they have put into the manuscript. It is sometimes hard to judge the accuracy of a result without a specific reference to all the corrections employed in a work, so I am very happy to see that the authors already considered almost all my comments beforehand.

I will admit that my ultimate hope was that any of the corrections that I pointed out would have been the one to actually allow the authors to reach close(r) to the experimental values and I thank the authors for spending that much effort to chase this down, even if it was ultimately not as successful as hoped.

Reviewer #1

Comments: I appreciate the authors' detailed response to comments. Listed below are other concerns that need to be addressed before publication.

Response: We thank the reviewer for the thoughtful comments. The comments were greatly helpful to further improve the quality of our work. Below, we provide point-by-point responses and revised work.

1. Response to Comment 1. It is important that the five differences that the authors state between the earlier paper (Nature communications 10.1 (2019): 1-12.) and current work be summarized in the introduction section of the manuscript. Otherwise, the current manuscript seems quite similar to the previous one.

[Response]:

We are thankful for the reviewer's constructive comments. Following the reviewer's suggestion, we included the additional discussion regarding the uniqueness of the current work differentiated from our previous work in the revised manuscript as follows.

Added text:

In the present work, we have developed a new class of heterostructured catalyst comprising perovskite oxide ($\text{La}_{0.5}\text{Sr}_{0.5}\text{CoO}_{3-\delta}$, LSC) and potassium ion-bonded molybdenum diselenide (K-MoSe_2) as the bifunctional catalysts for overall water electrolysis. The semiconducting 2H-phase MoSe_2 was moderately converted to metallic 1T- MoSe_2 via charge transfer from potassium atoms during the potassium metal intercalation process²⁶. In a previous report, Park et al. reported a heterostructured water electrolysis catalyst structure comprising perovskite oxide and transition metal dichalcogenides (TMDs) featured with unidirectional charge transfer enabling the in-situ local phase-transition in TMDs. Differently, the LSC/ K-MoSe_2 system in this study characterizes multidirectional charge transfer phenomenon, involving two-way charge transfer from K to MoSe_2 and from LSC to MoSe_2 , which led to significantly improved water electrolysis performance and operational stability.

2. Response to Comment 2. OK.

[Response]:

We thank very much the positive response from the reviewer on our Response to Comment 2.

3. Response to Comment 3. The XPS data is quite convincing that there is still molybdenum selenide after 2500 hours of testing. However, it would be helpful to show the XPS data for before and after electrolysis in the same chart.

Only oxides and oxidized species are possible under OER conditions. Reduced state of Mo and Se are not tenable at these potentials. In the added text, please modify the statement " MoSe_2 may potentially be decomposed into Mo and Se at the potentials of the OER electrode ..." to " MoSe_2 may potentially be decomposed into molybdenum oxides and selenate at the potentials of the OER electrode"

[Response]:

We appreciate the reviewer for providing the insightful comments. The reviewer's comments can be summarized into the followings.

1. Compiling the Mo 3d and Se 3d XPS data before and after 2,500 h stability test into one figure
2. Correcting the statement regarding the potential decomposition of MoSe_2 under OER conditions.

We agree that compiling the Mo 3d and Se 3d XPS data before and after the 2,500 h operational stability test into the same chart is more helpful for the readers and that the proposed description regarding the potential decomposition of MoSe_2 in the OER state is more appropriate. Following the reviewer's suggestion, we revised the related Figure and context in Supplementary Information and revised manuscript as follows.

Original figure:

Supplementary Figure 25 High resolution XPS spectra of Mo 3d and Se 3d for LSC/K-MoSe₂ in OER condition after 2,500 h of chronopotentiometric stability test.

Revised figure:

Supplementary Figure 26 High resolution XPS spectra of a, c Mo 3d and b, d Se 3d for LSC/K-MoSe₂ in OER condition before and after 2,500 h of chronopotentiometric stability test.

Original text:

MoSe₂ may potentially be decomposed into Mo and Se at the potentials of the OER electrode during prolonged electrolysis⁶⁰. To verify the stability of MoSe₂ in LSC/K-MoSe₂, we performed XPS analysis for Mo and Se in LSC/K-MoSe₂ on the OER electrode after 2,500 h of chronopotentiometric stability test. Although some partial oxidation was observed, the integrity of MoSe₂ was well preserved without decomposition (**Supplementary Figure 25** and **Table 9**).

Revised text:

MoSe₂ may potentially be decomposed into **molybdenum oxides and selenate** at the potentials of the OER electrode during prolonged electrolysis⁶⁰. To verify the stability of MoSe₂ in LSC/K-MoSe₂, we performed XPS analysis for Mo and Se in LSC/K-MoSe₂ on the OER electrode after 2,500 h of

chronopotentiometric stability test. Although some partial oxidization was observed, the integrity of MoSe₂ was well preserved without decomposition (**Supplementary Figure 26 and Table 9**).

4. Response to Comment 4. The enhancement of the electrode kinetics by molybdenum is central to the conclusions in this manuscript. The speculation about how the Mo increases the OH and OOH adsorption is quite plausible, as the rate-determining step involves the oxidation of adsorbed intermediates. However, the reviewer cannot find any experimental basis for this proposed mechanism. For example, have the authors found differences in Tafel Slope, lowering of adsorption energy of the surface intermediates, or the shifts in binding energy of cobalt in XPS that support this hypothesis? It would be important to provide some discussion on this in addition to speculating the mechanism. For guidance, please see: Malkhandi, S., et al. "Design insights for tuning the electrocatalytic activity of perovskite oxides for the oxygen evolution reaction." *The Journal of Physical Chemistry C* 119.15 (2015): 8004-8013.

[Response]:

We thank the reviewer for the valuable comments. We carefully reviewed the paper mentioned by the reviewer (S. Malkhandi et al., Design insights for tuning the electrocatalytic activity of perovskite oxides for the oxygen evolution reaction. *J. Phys. Chem. C*, 119, 8004-8013 (2015)) regarding the performance of the water electrolysis catalysts. We agree with the reviewer's comment that the comparative analysis (experimental and computational) between LSC/K-MoSe₂ and LSC should be well structured in the study since the molybdenum in the proposed heterostructure has a significant influence on the catalytic activity. In fact, we have included the experimental and computational analysis results (e.g., Tafel slope, shift in binding energy of cobalt in XPS, free energy barrier, and density of states) to elucidate the origin of the performance enhancement in LSC/K-MoSe₂ compared to that of LSC in the original manuscript and Supplementary Information as follows.

- Tafel slope (**Figure 2e**)
- The shifts in binding energy of cobalt in XPS (**Supplementary Figure 14**)
- Free energy barrier and density of states (**Figure 4f–h, Supplementary Figures 20–22**)

Following the reviewer's suggestion, we included additional discussion describing the effect of molybdenum to the enhanced water electrolysis performance of LSC/K-MoSe₂ compared to that of LSC in Supplementary Information.

Added text:

Thus, the origin of OER performance enhancement of LSC in the heterostructure is not attributed to the doping effect of molybdenum or its oxide derivatives. In our system, Mo, rather than serving as the main contributor for the OER active site, is believed to participate in the continuous charge transfer process with LSC during the OER reaction, which enhances the electrophilicity of LSC, leading to the increased adsorption of OER intermediates such as OH* and OOH*, and modulates the electronic structure of LSC/K-MoSe₂ favorable for OER process. **The improvement in OER kinetics of LSC/K-MoSe₂ compared to that of LSC owing to the presence of Mo in the OER environment was also previously verified via experimental and computational analyses (low Tafel slope value (**Figure 2e**), upshift of XPS spectra of Co 2p peak (**Supplementary Figure 14**), and reduced free energy barrier and density of states (**Figure 4f–h, Supplementary Figures 21–23**)).**

Reviewer#2

Comments: I'm glad to see all the efforts the authors have put in to improve the manuscript. They have addressed all my comments and I recommend its publication in Nature Comm.

Response: We sincerely appreciate the reviewer for the thoughtful and positive comments once again.

Reviewer#3

Comments: First of all I want to thank the authors for taking the time to discuss and address the comments made by all reviewers in detail. I think the changes made to the manuscript significantly strengthened it and I would at this point fully support a publication in Nat Comm. I had some minor comments left in regards to some of the answers the authors gave, which I will add below. I will leave it to the discretion of the authors on what level they address these.

Response: We sincerely appreciate the reviewer for providing the positive and valuable comments to strengthen the quality of our manuscript. The detailed response to the comments of the reviewer is provided below.

1. For me personally, at this point, the best reported onset potentials for the OER on IrO₂ would be around 1.53V by Goddard and coworkers (JACS 2016) by considering an OO- type structure, not completely dissimilar to the one discussed here for the studied catalyst. Similar structures were also reported for RuO₂ by Rossmeisl(PCCP, 2014, DFT only) and then later Rossmeisl and Shao-Horn (EES 2017, DFT&Experiment) on RuO₂.

Now it can be argued if the results shown there are really convincing, however, there are at least some indications that DFT and experiment for these systems can be brought to overlap more closely.

It might be interesting to discuss the current results in relation to these results and what trends could be discerned from them in relation to the current results, but I would see this as optional.

[Response]:

We appreciate the reviewer's valuable opinion and positive comments on our revised work. We added some discussion in Supplementary Information related to the issue mentioned by the reviewer.

Added text:

Supplementary Note 3

The solvation corrected free energy barriers in the rate determining step for OER in LSC/K-MoSe₂ (2.13 eV) obtained using VASPsol code showed an increased discrepancy from the experimentally measured value (1.46 eV @ 10 mA cm⁻²) than the non-corrected result, due to the lack of information on the inhomogeneous permittivity near the interfaces. The simulation of exact solvation effect is known to be extremely challenging even in the "implicit + explicit" solvation model since the computed interface properties are sensitive to the simulation parameters and the structure of water molecules³¹⁻³³. In addition, the assumption in the detailed OER process should be one of the reasons for the discrepancy between experiment and simulation. Although we evaluated the free energy barriers based on the LOM elementary reactions as described above, there can be alternative elementary reaction paths which are more favorable, and each of the elementary reaction can be divided into unknown fundamental reactions that may affect the simulation results. While clarifying the most suitable elementary reactions pathway is certainly worthwhile and considered as a successful approach to achieve an agreed result with experiments³⁴, it might be beyond the scope of this study

Added reference:

Supplementary References

34. Ping, Y. et al. The Reaction Mechanism with Free Energy Barriers at Constant Potentials for the Oxygen Evolution Reaction at the IrO₂ (110) Surface. *J. Am. Chem. Soc.* **139**, 149-155 (2017).

2. While I strongly appreciate the authors re-relaxing the structure for this intermediate with a stricter criterion and the following reduction in the ΔG^* to 0.11eV, I want to ask them consider to move this discussion purely to the SI and leave the initial value in the main text. The reason for this request is simply that the chosen relaxation criterion in this single case makes the results not 100% comparable to all the other results in the study, which can lead to inconsistencies. However, I find the result very positive in relation to the initial consideration in my question and I thank the authors for taking the time to recalculate this.

[Response]:

We appreciate the reviewer's helpful suggestion, and we also agree with the reviewer's opinion. As suggested, we moved the reduced ΔG_{H^*} results and discussion to Supplementary Information, leaving the original results in the main text.

Original text:

The LSC/K-MoSe₂ structures are suitable catalysts for providing the most favorable HER environment for H₂ production with the smallest ΔG_{H^*} of 0.11 eV, whereas the ΔG_{H^*} values for LSC, 2H-MoSe₂, and K-MoSe₂ are 2.49, 2.09, and 1.22 eV, respectively. The optimal ΔG_{H^*} in LSC/K-MoSe₂ is associated with the increased number of states near the energy of normal hydrogen electrode (NHE) potential (see **Supplementary Figure 17**), thus increasing the interactions between the hydrogen *s* orbital and 1T-MoSe₂ states.

Revised text:

The LSC/K-MoSe₂ structures are suitable catalysts for providing the most favorable HER environment for H₂ production with the smallest ΔG_{H^*} of 0.25 eV, whereas the ΔG_{H^*} values for LSC, 2H-MoSe₂, and K-MoSe₂ are 2.01, 1.44, and 0.69 eV, respectively. The optimal near-zero ΔG_{H^*} in LSC/K-MoSe₂ is associated with the increased number of states near the energy of normal hydrogen electrode (NHE) potential (see **Supplementary Figure 17**), thus increasing the interactions between the hydrogen *s* orbital and the 1T-MoSe₂ states.

Original figure:

Fig. 4 Complementary charge transfer phenomena in bifunctional LSC/K-MoSe₂ catalysts. **e** Free energy diagrams for HER for LSC, 2H-MoSe₂, K-MoSe₂, and LSC/K-MoSe₂.

Revised figure:

Fig. 4 Complementary charge transfer phenomena in bifunctional LSC/K-MoSe₂ catalysts. **e** Free energy diagrams for HER for LSC, 2H-MoSe₂, K-MoSe₂, and LSC/K-MoSe₂.

Added figure:

Supplementary Figure 18 Free energy diagrams with the smallest ΔG_{H^+} in the LSC/K-MoSe₂ for HER for LSC, 2H-MoSe₂, K-MoSe₂, and LSC/K-MoSe₂.

Added text:

Supplementary Note 2

Furthermore, the large number of states near the Fermi level in LSC/K-MoSe₂ can improve the electrical conductivity of the catalyst, enhancing the overall catalytic activity in LSC/K-MoSe₂ heterostructure with increased charge transfer kinetics.

The smallest ΔG_{H^+} in the LSC/K-MoSe₂ and the order of ΔG_{H^+} values (LSC/K-MoSe₂ < K-MoSe₂ < 2H-MoSe₂ < LSC) was also confirmed with a stricter force convergence criterion (5×10^{-3} eV/Å) where ΔG_{H^+} values for LSC, 2H-MoSe₂, K-MoSe₂, and LSC/K-MoSe₂ were 2.49, 2.09, 1.22, and 0.11 eV, respectively (**Supplementary Figure 18**).

3. I thank the authors for further specifying their methods section to reflect the true extent of effort and expertise they have put into the manuscript. It is sometimes hard to judge the accuracy of a result without a specific reference to all the corrections employed in a work, so I am very happy to see that the authors already considered almost all my comments beforehand.

I will admit that my ultimate hope was that any of the corrections that I pointed out would have been the one to actually allow the authors to reach close(r) to the experimental values and I thank the authors for spending that much effort to chase this down, even if it was ultimately not as successful as hoped.

[Response]:

We sincerely appreciate the reviewer for the thoughtful and positive comments once again.